# Activation mechanism and novel binding sites of the BK$_{Ca}$ channel activator CTIBD

Narasaem Lee[1],[*], Subin Kim[1],[*], Na Young Lee[1], Heeji Jo[1], Pyeonghwa Jeong[2], Haushabhau S Pagire[3], Suvarna H Pagire[3], Jin Hee Ahn[3], Mi Sun Jin[1] (ORCID), Chul-Seung Park[1] (ORCID)

The large-conductance calcium-activated potassium (BK$_{Ca}$) channel, which is crucial for urinary bladder smooth muscle relaxation, is a potential target for overactive bladder treatment. Our prior work unveiled CTIBD as a promising BK$_{Ca}$ channel activator, altering $V_{1/2}$ and $G_{max}$. This study investigates CTIBD's activation mechanism, revealing its independence from the Ca$^{2+}$ and membrane voltage sensing of the BK$_{Ca}$ channel. Cryo-electron microscopy disclosed that two CTIBD molecules bind to hydrophobic regions on the extracellular side of the lipid bilayer. Key residues (W22, W203, and F266) are important for CTIBD binding, and their replacement with alanine reduces CTIBD-mediated channel activation. The triple-mutant (W22A/W203A/F266A) channel showed the smallest $V_{1/2}$ shift with a minimal impact on activation and deactivation kinetics by CTIBD. At the single-channel level, CTIBD treatment was much less effective at increasing $P_o$ in the triple mutant, mainly because of a drastically increased dissociation rate compared with the WT. These findings highlight CTIBD's mechanism, offering crucial insights for developing small-molecule treatments for BK$_{Ca}$-related pathophysiological conditions.

## Introduction

Activation of the large-conductance Ca$^{2+}$-activated K$^+$ channel (BK$_{Ca}$ channel; also called Maxi K or KCa1.1) occurs through depolarization of the membrane voltage and an increase in the intracellular Ca$^{2+}$ concentration (Latorre & Miller, 1983; Golowasch et al, 1986; Latorre et al, 1989). When the BK$_{Ca}$ channel is open, K$^+$ ions quickly exit the cell through the channel at physiological K$^+$ gradients and membrane potentials, causing the membrane to become hyperpolarized (Lancaster & Nicoll, 1987; Storm, 1987). BK$_{Ca}$ channels are composed of a tetramer of the pore-forming α-subunit (*Slo1*), sometimes with auxiliary β- and γ-subunits and LINGO family (Brenner et al, 2000a;

Orio et al, 2002; Zhang & Yan, 2014; Dudem et al, 2020). The BK$_{Ca}$ channel is fully functional without β- and γ-subunits, which may change the channel characteristics such as sensitivities for Ca$^{2+}$, voltage, and agonists (Orio & Latorre, 2005; Guan et al, 2017; Gonzalez-Perez & Lingle, 2019). The α-subunit of the BK$_{Ca}$ channel has seven transmembrane domains, S0–S6, and a large intracellular domain that contains two regulators of conductance for K$^+$ (RCK) domains (Lee & Cui, 2010; Yuan et al, 2010). The voltage-sensing residues are in the S1–S4 transmembrane domains, and the S5–S6 transmembrane domains comprise the pore of the channel (Meera et al, 1997; Yellen, 2002; Ma et al, 2006). The Ca$^{2+}$ binding sites in the RCK1 and RCK2 domains sense the intracellular Ca$^{2+}$ concentration (Schreiber & Salkoff, 1997; Xia et al, 2002; Moczydlowski, 2004). Truncated BK$_{Ca}$ channels lacking the large intracellular domain are not activated by an increase in Ca$^{2+}$ because the RCK1 and RCK2 domains that sense intracellular Ca$^{2+}$ are absent (Budelli et al, 2013). The gating mechanism of the BK$_{Ca}$ channel is regulated by pore gating, Ca$^{2+}$ sensing, voltage sensing, and their couplings (Horrigan & Aldrich, 2002).

Expressed in various human tissues, including the brain, bladder, heart, and smooth muscle (Dworetzky et al, 1994; McCobb et al, 1995; Brenner et al, 2000b; Poulsen et al, 2009), BK$_{Ca}$ channels regulate the contraction of smooth muscle (Fernandes et al, 2015), neurotransmitter release (Wang, 2008), hormone secretion (Wang et al, 1994), and arterial tone (Löhn et al, 2001). Thus, BK$_{Ca}$ channels are associated with stroke (Gribkoff et al, 2001), erectile dysfunction (Werner et al, 2005), cerebellar ataxia (Srinivasan et al, 2022), and overactive bladder (OAB) (Meredith et al, 2004). BK$_{Ca}$ channels are highly expressed in bladder smooth muscle cells (Chen & Petkov, 2009). Activation of BK$_{Ca}$ channels causes hyperpolarization of the cell membrane and relaxes the phasic contraction of bladder smooth muscle (Herrera et al, 2000). BK$_{Ca}$ channel knockout mice have an OAB and urinary incontinence because the bladder smooth muscle is not controlled normally (Meredith et al, 2004), and a decrease in BK$_{Ca}$ channel expression is associated with patients with an overactive detrusor (Chang et al, 2010; Hristov et al, 2013). Although several BK$_{Ca}$ channel agonists have been developed to

---

[1]School of Life Sciences, Gwangju Institute of Science and Technology (GIST), Gwangju, Republic of Korea   [2]Department of Chemistry, Duke University, Durham, NC, USA
[3]Department of Chemistry, Gwangju Institute of Science and Technology (GIST), Gwangju, Republic of Korea

Correspondence: misunjin@gist.ac.kr; cspark@gist.ac.kr
Heeji Jo's present address is R&D Planning Department, SD Biosensor, Inc., Suwon-si, Republic of Korea
*Narasaem Lee and Subin Kim contributed equally to this work

---

target OAB (Layne et al, 2010; Soder & Petkov, 2011; Nausch et al, 2014), none have passed clinical trials because of low efficacy and a lack of significant physiological effects (Oger et al, 2011; Bentzen et al, 2014).

We previously reported a novel $BK_{Ca}$ channel activator, CTIBD, based on a 4-phenylisoxazol-5-yl benzene skeleton (Lee et al, 2021), that shifts the G-V relationship in the negative direction and decreases the half-activation voltage ($V_{1/2}$) by ~50 mV at a concentration of 10 $\mu$M. CTIBD activates $BK_{Ca}$ channels that coexpress $\beta$1- and $\beta$4-subunits, which are expressed primarily in bladder smooth muscle (Wu & Marx, 2010), induces relaxation of rat bladder smooth muscle strips, and decreases the number of times rats void when it is administered intraperitoneally. A cell-based fluorescence assay demonstrated that CTIBD induces greater $BK_{Ca}$ channel activation than that induced by other well-known $BK_{Ca}$ channel activators such as NS1619, NS11021, and rottlerin (Lee et al, 2021). In addition, there are several $BK_{Ca}$ channel activators such as Cym04, omega-3 docosahexaenoic acid, GoSlo family, and BC5, which can be referred to for understanding the $BK_{Ca}$ channel activation mechanism of CTIBD (Gessner et al, 2012; Hoshi et al, 2013; Webb et al, 2015; Zhang et al, 2022). Thus, CTIBD activates $BK_{Ca}$ channels and effectively relaxes bladder smooth muscle.

As a follow-up to a previous study, we investigated the molecular mechanism of $BK_{Ca}$ channel activation by CTIBD. We first determined the $BK_{Ca}$ channel gating component(s) affected by CTIBD. Using truncated $BK_{Ca}$ channels lacking the cytosolic $Ca^{2+}$-sensing domains and recording the current of WT $BK_{Ca}$ channels at extreme negative membrane voltages, we demonstrated that the $Ca^{2+}$- and voltage-sensing mechanisms are not required for CTIBD-dependent channel activation. These results further indicated that CTIBD might affect the intrinsic gating that occurs primarily within the transmembrane domains, including the ion-conduction pore (Horrigan & Aldrich, 2002; Budelli et al, 2013; Rockman et al, 2020). In addition, we show the cryo-electron microscopy (cryo-EM) structure of homomeric human $BK_{Ca}$ channel $\alpha$-subunits (hSlo1) in complex with CTIBD. Based on the structural information, we identified the major contributing residues involved in protein–CTIBD interactions. When these residues were replaced with alanine (Ala), the activation effects of CTIBD were significantly compromised. Among the various mutants, the triple mutant (W22A/W203A/F266A) showed the smallest $V_{1/2}$ shift in response to CTIBD, validating the importance of these residues in CTIBD-dependent channel activation.

# Results

## $Ca^{2+}$ and voltage sensing are not essential for CTIBD-mediated $BK_{Ca}$ channel activation

$BK_{Ca}$ channel activation is affected by the intracellular $Ca^{2+}$ ion concentration and membrane voltage. Using rSlo1-Kv-minT channels, in which the entire cytosolic C-terminus of the rat $BK_{Ca}$ channel $\alpha$-subunit containing the $Ca^{2+}$-sensing domains was replaced with 11 amino acid residues from the tail of Kv1.4 (Budelli et al, 2013), we measured $Ca^{2+}$ sensing and CTIBD-mediated $BK_{Ca}$ channel activation (Fig 1A). When only the nuclease-free water

without mRNA was injected into the oocytes, small leak currents were observed (Fig 1B). Based on this fact, it can be inferred that the currents recorded from oocytes injected with rSlo1-Kv-minT mRNA were due to the opening of the modified channels expressed on the membrane. When 3 $\mu$M CTIBD was added to the truncated $BK_{Ca}$ channel on the extracellular side of the membrane, we observed an increase in the ionic current and conductance in the absence of intracellular $Ca^{2+}$ (Fig 1B and C). Each data point in the G-V curve was normalized to the conductance value obtained after treatment with the vehicle at 250 mV. It is worth noting that the maximum conductance could not be identified because of membrane instability at high membrane voltages and the $Ca^{2+}$ insensitivity of the rSlo1-Kv-minT mutant channel. Compared with the vehicle control, 3 $\mu$M CTIBD with no intracellular $Ca^{2+}$ increased the conductance significantly at 190, 210, and 230 mV by 2.2-, 2.2-, and 2.6-fold, respectively. Similar results were obtained with 30 $\mu$M intracellular $Ca^{2+}$ (Fig 1D). Thus, the $Ca^{2+}$-sensing domain is not essential for CTIBD-mediated $BK_{Ca}$ channel activation.

Because the $BK_{Ca}$ channel is a voltage-gated $K^+$ channel activated by membrane depolarization, we determined the relationship between voltage sensing and CTIBD-mediated $BK_{Ca}$ channel activation. We determined the effects of CTIBD on $BK_{Ca}$ channels at fixed negative membrane voltages (−90 to −60 mV) with no intracellular $Ca^{2+}$. The voltage-sensing domains of the channel were mainly in the resting, deactivated state under these experimental conditions (Rockman et al, 2020). Because the open-state probability of the channel is extremely low under these conditions, we performed patch-clamp experiments with between 27 and 60 active channels on a single membrane patch (Horrigan & Aldrich, 2002; Wang et al, 2009). When 10 $\mu$M CTIBD was added to the membrane from the extracellular side, the number of opening events increased significantly (Fig 2A and B). Upon treatment with CTIBD, some long opening events, which were not found in vehicle-treated channels, were also observed within the same time periods (Fig 2C and D). In the absence of CTIBD, the open-state probabilities were estimated as $1.56 \pm 0.20 \times 10^{-5}$, $1.05 \pm 0.13 \times 10^{-5}$, $7.46 \pm 0.68 \times 10^{-6}$, and $5.13 \pm 0.18 \times 10^{-6}$ at −60, −70, −80, and −90 mV, respectively. However, 10 $\mu$M CTIBD increased the open-state probabilities dramatically to $1.61 \pm 0.27 \times 10^{-3}$ (103-fold), $1.01 \pm 0.12 \times 10^{-3}$ (96-fold), $4.88 \pm 0.43 \times 10^{-4}$ (65-fold), and $3.36 \pm 0.29 \times 10^{-4}$ (65-fold), respectively (Fig 2E). Thus, CTIBD activates $BK_{Ca}$ channels even at very negative voltages, at which the voltage sensors are in a resting conformation.

Because neither voltage sensing nor $Ca^{2+}$ sensing by the membrane is required for CTIBD-dependent channel activation, we aimed to determine whether CTIBD alters the intrinsic gating equilibrium of the channel. The binding of CTIBD to the transmembrane domain of the channel may stabilize its activated (or open) conformation.

## Structural identification of the CTIBD binding sites

To investigate the activation mechanism of CTIBD at the molecular level, we determined the cryo-EM structure of the detergent-purified human $BK_{Ca}$ channel in complex with CTIBD. The final map was obtained from the best three-dimensional (3D) class containing 288,878 particles and yielded resolutions of 4.2 and 3.9 Å with C1 and C4 symmetry, respectively (Figs S1A–C and S2).

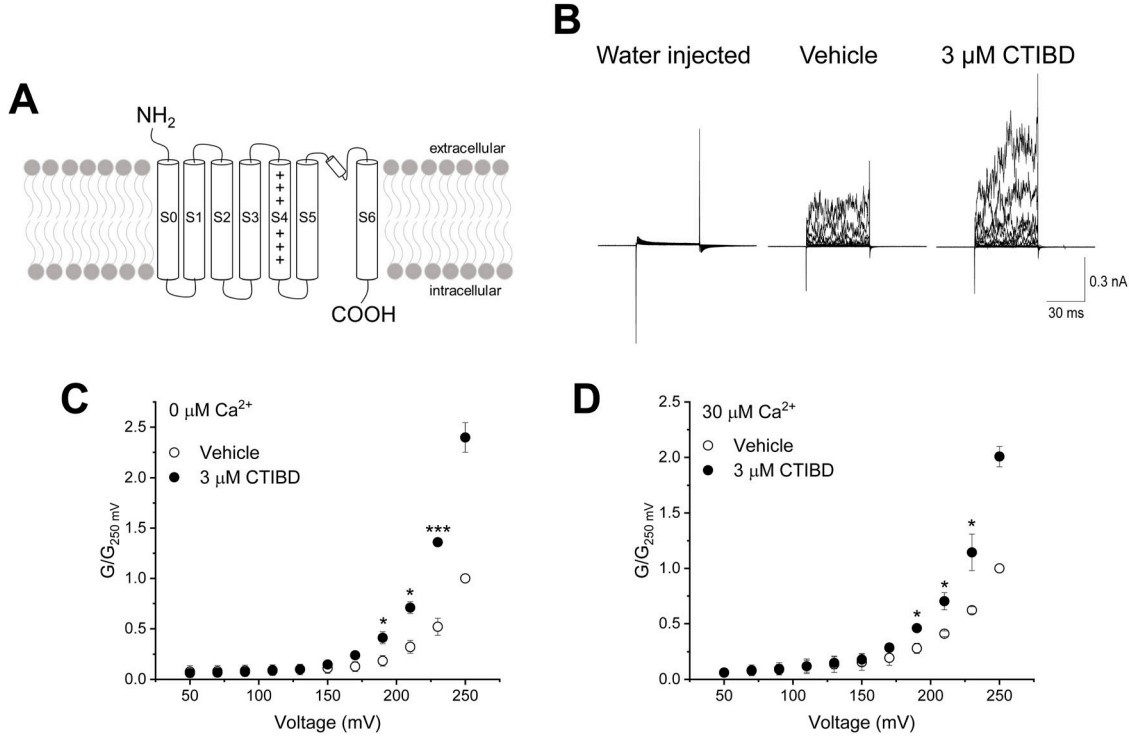

**Figure 1. Effects of CTIBD on rSlo1-Kv-minT.**
**(A)** Topology of the rSlo1-Kv-minT mutant channel. The C-terminal domain of the BK$_{Ca}$ channel was replaced with 11 amino acid residues from the tail of Kv1.4 (Budelli et al, 2013). **(B)** Representative current traces of rSlo1-Kv-minT with 0 $\mu$M intracellular Ca$^{2+}$. The vehicle was 0.1% DMSO. CTIBD was applied to the extracellular side of the channel with 50-ms voltage pulses. Currents were recorded for each of the voltage pulses, which increased from −30 to +250 mV in 20-mV increments. The holding voltage was −100 mV. **(C, D)** Conductance–voltage relationship of rSlo1-Kv-minT with 0 $\mu$M (C) and 30 $\mu$M (D) intracellular Ca$^{2+}$. After voltage pulse initiation, the average conductance was obtained from the outward current between 35 and 45 ms. All currents were normalized to the conductance of the vehicle-treated channel at 250 mV. Data are presented as the mean ± SEM. *$P < 0.05$, ***$P < 0.001$ (two-sample t test, n = 3).

Comparison of the CTIBD-bound structure with the apo BK$_{Ca}$ tetramer (Ca$^{2+}$-bound, open state, Protein Data Bank [PDB] ID 6V38) (Tao & MacKinnon, 2019) revealed their nearly identical overall folds, with a C$\alpha$ root mean square deviation of 1.8 Å, and similar radius of the ion-conduction pathway (Fig S3A–C) (Smart et al, 1996). Despite this similarity, two clear and distinct densities corresponding to the planar molecular structure of CTIBD were observed, positioned within an ~5 Å proximity of each other (Fig 3A and B). The first CTIBD molecule (CTIBD1) occupies the hydrophobic cavity formed by residues from the S0 and S5 helices (Fig 3C and D). Conversely, the binding site of the second CTIBD molecule (CTIBD2) remains fully exposed to the extracellular membrane face, yet remains anchored to the channel via a phi-stacking interaction between the 1,3-benzenediol ring of CTIBD2 and the highly conserved Trp263 (W263) of the turret loop (Fig 3C–E). The van der Waals interactions between CTIBD2 and the adjacent cholesteryl hemisuccinate (CHS) also contribute to the binding free energies. In the structure, residues R20 and P262 do not directly participate in CTIBD binding (Fig 3C–E). However, they are located immediately behind F266 and W263, respectively, and play a role in fixing the conformation of their side chains. This binding mode of CTIBD differs from those of inhibitors that bind to the *Drosophila* BK$_{Ca}$ channel, such as the fungal neurotoxin verruculogen and the anthelmintic drug emodepside (Raisch et al, 2021). Specifically, the verruculogen

binding site is located at the interface between the S5 and S6 segments of the channel, whereas the emodepside binding site is located in the S6 transmembrane segment of the channel's pore-forming $\alpha$-subunit. Furthermore, our study revealed differences in the bound lipid species between the apo and CTIBD-bound structures. In the apo state, one cholesterol and three molecules of 1-palmitoyl-2-oleoyl-sn-glycero-3-phosphocholine (POPC) occupy the interface between two adjacent subunits (Fig S4A). In our structure, however, CTIBD and CHS replace the cholesterol and POPCs, respectively (Fig 3C). Consequently, to avoid steric collision with CHS, the indole moiety of W275 is flipped by ~120° relative to the orientation observed in the apo state (Fig S4B). Collectively, our findings offer valuable insights into the interaction mechanism and binding mode of CTIBD, paving the way for the design of a more potent BK$_{Ca}$ channel activator.

### Effects of CTIBD on Ala mutant BK$_{Ca}$ channels

Two CTIBD molecules mainly interact with eight amino acid residues on the BK$_{Ca}$ channel, and these amino acids are conserved in both human and rat BK$_{Ca}$ channels (Fig 3D and E). Therefore, we constructed eight single-mutant channels with Ala substitutions individually and measured the effects of the mutations on CTIBD efficacy. All mutant channels were expressed well on the oocyte

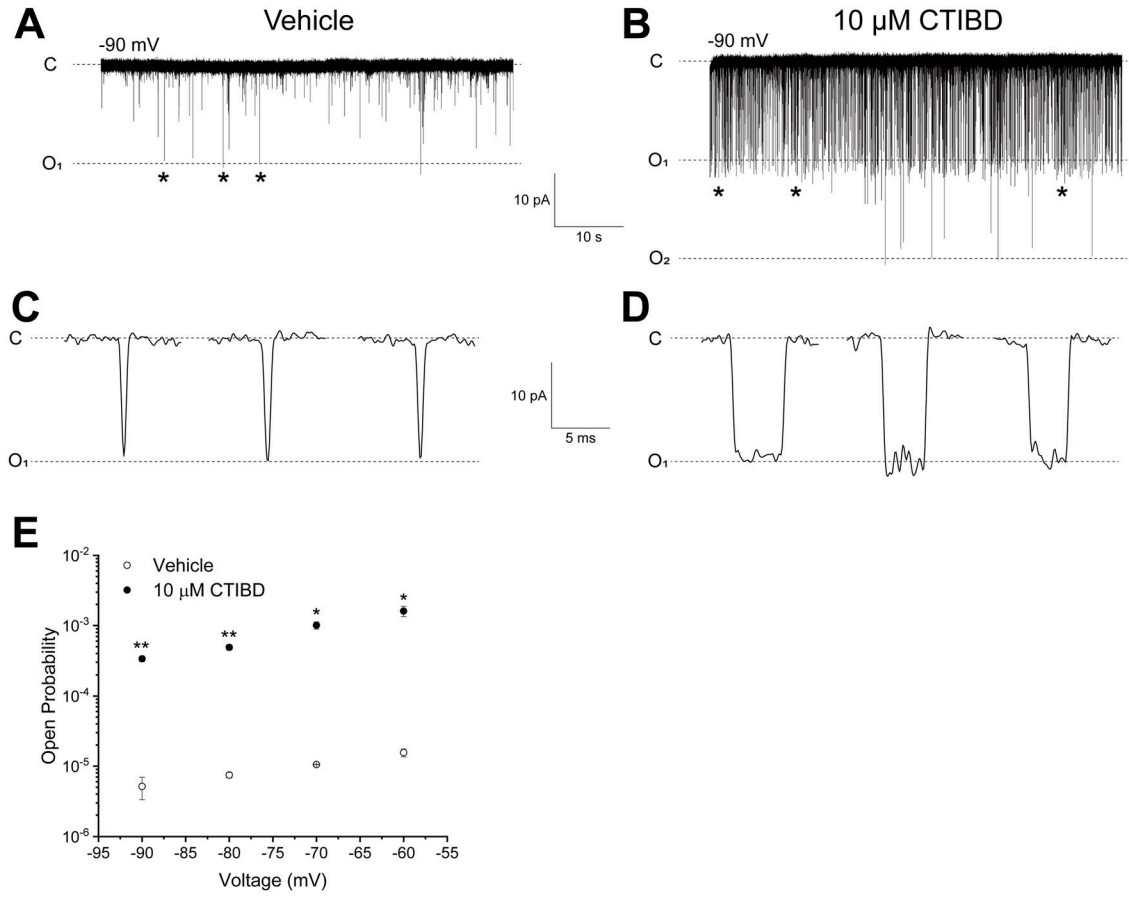

**Figure 2. Effect of CTIBD on the $P_o$ of the $BK_{Ca}$ channel at negative membrane voltages.**
**(A, B)** Representative current traces of the $BK_{Ca}$ channel with the vehicle (A) and 10 $\mu$M CTIBD (B). The intracellular $Ca^{2+}$ concentration was 0 $\mu$M. The vehicle was 0.1% DMSO. CTIBD was applied to the extracellular side of the channel. The number of channels on the patch membrane was 45, which was calculated by dividing the maximum current level at ≥150 mV by the single-channel conductance. **(C, D)** Representative channel opening events with the vehicle (C) or 10 $\mu$M CTIBD (D) are shown. **(A, B)** Position of each opening is indicated by an asterisk (*) on the representative traces (A, B). **(E)** $P_o$ of $BK_{Ca}$ channels treated with the vehicle or 10 $\mu$M CTIBD. Data are presented as the mean ± SEM. *$P < 0.05$, **$P < 0.01$ (two-sample $t$ test, n = 3).

membrane, and their macroscopic currents were measured successfully. The effects of the Ala mutations on CTIBD-dependent channel activation were estimated by measuring the shift of $V_{1/2}$ values (Fig S5 and Table S1). Because Ala substitutions at three residues, Trp22, Trp203, and Phe266, comprising the CTIBD1 site reduced the $V_{1/2}$ shift most significantly, we constructed a triple-mutant channel (W22A/W203A/F266A) and investigated the effects of CTIBD in detail. Treatment with 10 $\mu$M CTIBD produced much smaller tail currents in the triple-mutant $BK_{Ca}$ channel than those in the WT $BK_{Ca}$ channel (Fig 4A and B). Although the $G$-$V$ curve of the triple-mutant channel was shifted in a negative direction by 10 $\mu$M CTIBD, this effect was much smaller than that in the WT $BK_{Ca}$ channel (Fig 4C). Although treatment of the WT $BK_{Ca}$ channel with 10 $\mu$M CTIBD decreased the $V_{1/2}$ by 86.4 ± 2.8 mV, from 103.1 ± 7.4 to 16.7 ± 5.2 mV, the same concentration of CTIBD decreased the $V_{1/2}$ value of the triple mutant by only 12.8 ± 1.5 mV, from 128.3 ± 1.7 to 115.5 ± 2.1 mV. All other Ala mutants showed significantly smaller negative shifts of the $G$-$V$ relationship and thus smaller negative $\Delta V_{1/2}$ values than that of the WT channel (Figs 4D and S5).

### Effects of CTIBD on the gating kinetics of WT and triple-mutant $BK_{Ca}$ channels

We found that the peak of the tail currents produced by the termination of the depolarization pulse in the WT and triple-mutant (W22A/W203A/F266A) $BK_{Ca}$ channels differed substantially. Therefore, we further investigated the effects of CTIBD on the gating kinetics of WT and mutant $BK_{Ca}$ channels. We determined the time constants for activation (opening) and deactivation (closing) for various membrane voltages. Voltage-clamp pulses were applied from –80 to 180 mV in increments of 10 mV while holding the resting potential at –100 mV. Upon the application of a 150-mV voltage-clamp pulse, vehicle-treated and 10 $\mu$M CTIBD-treated WT $BK_{Ca}$ channels exhibited a clear difference in time constants, whereas the triple-mutant $BK_{Ca}$ channel did not (Fig 5A and B). The activation and deactivation time constants were obtained by fitting current traces at each voltage as a single-exponential function (Fig 5C and D). With the application of 10 $\mu$M CTIBD, the activation time constant of the WT $BK_{Ca}$ channel decreased significantly at 150, 160, and 170 mV, whereas significant differences were observed only at 180

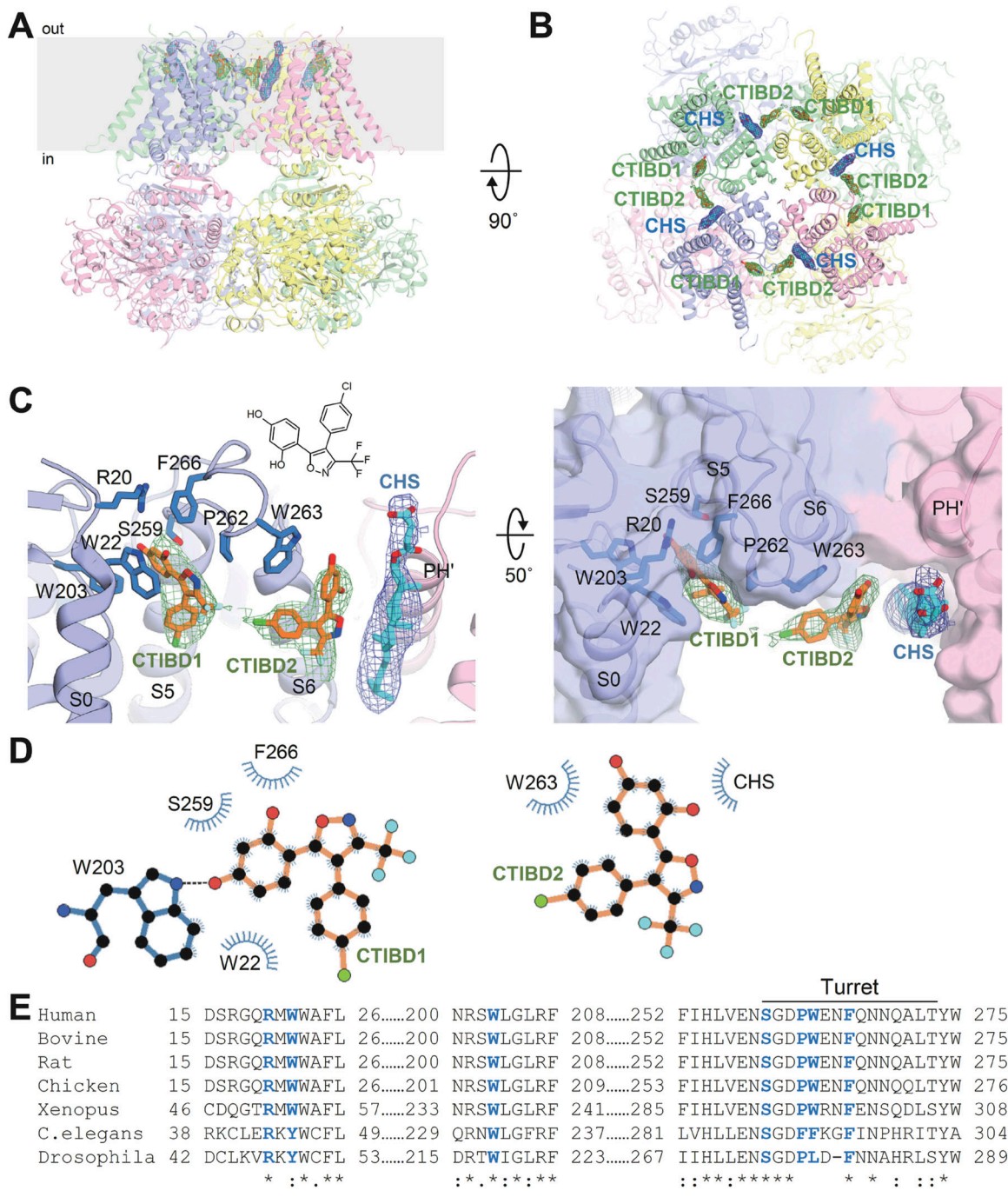

**Figure 3. Cryo-EM structure of the BK$_{Ca}$ channel in complex with CTIBD.**
**(A, B)** Overall structure of the human BK$_{Ca}$-CTIBD complex. The bound CTIBD and cholesteryl hemisuccinate are shown as sticks. The cryo-EM maps of CTIBD (green mesh) and cholesteryl hemisuccinate (blue mesh) are contoured at the 3σ level. Each subunit is colored in blue, green, pink, or yellow. **(A, B)** Side view (A) and a top view (B) are shown. **(C)** Close-up views of the CTIBD binding site from within the plane of the membrane (left) and top (right). Bound CTIBD and key interacting residues are represented by sticks and are colored according to heteroatoms. **(D)** Interactions between the BK$_{Ca}$ channel and CTIBD were analyzed using LigPlot+ software (Laskowski & Swindells, 2011; Wallace et al, 1995). Residues involved in nonpolar and van der Waals interactions within a distance of 4 Å are depicted as blue semicircles. **(E)** Sequence alignment of the CTIBD-interacting residues. The conserved residues are highlighted in blue.

and 190 mV for the triple mutant. With the application of 10 μM CTIBD, the activation time constant of the WT BK$_{Ca}$ channel was ~50% smaller at 150 mV (Table S2). The effects of CTIBD were more dramatic for deactivation or closing of the channel. Although the deactivation time constant of the WT BK$_{Ca}$ channel increased by 7.1-fold at 150 mV with the application of 10 μM CTIBD, that of the triple-mutant BK$_{Ca}$ channel was increased only 1.6-fold under identical experimental conditions (Table S3), indicating that triple-mutant

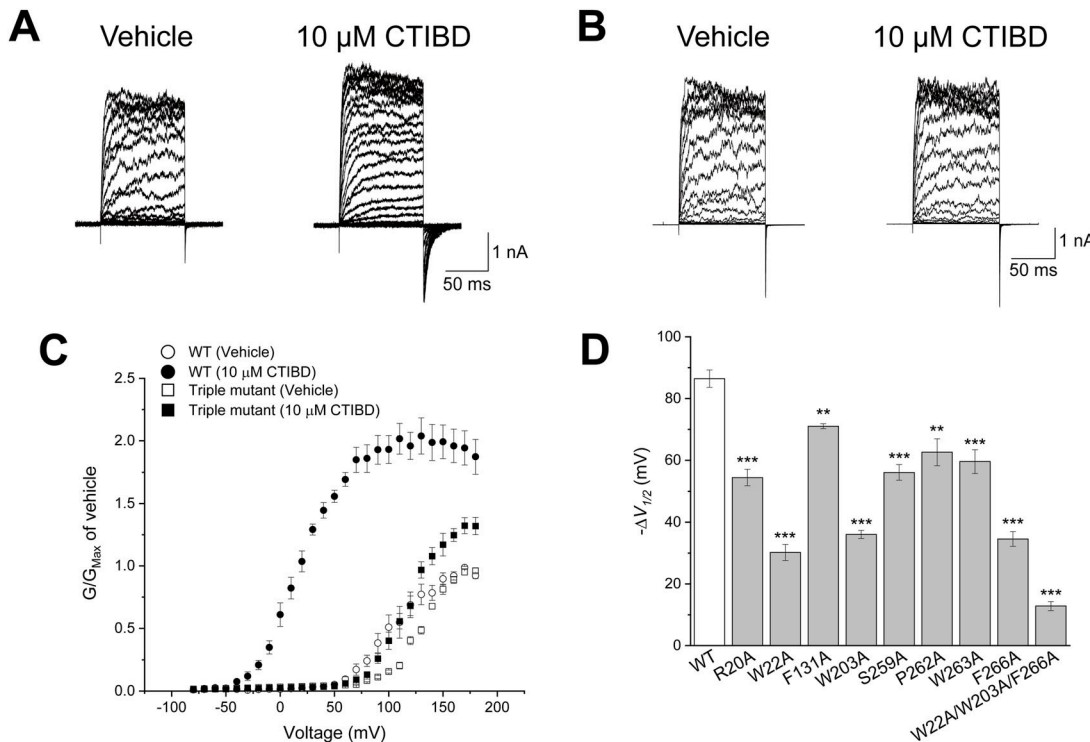

**Figure 4. Effects of CTIBD on Ala mutant BK_Ca channels.**
**(A, B)** Representative current traces of the WT (A) and triple-mutant (W22A/W203A/F266A) BK_Ca channels (B). The intracellular $Ca^{2+}$ concentration was 3 $\mu$M. The vehicle was 0.1% DMSO. CTIBD was applied to the extracellular side of the channel. The duration of the voltage pulses was 100 ms. Currents were recorded for each of the voltage pulses, which increased from −80 to 180 mV in 10-mV increments. The holding voltage was −100 mV. **(C)** Conductance–voltage relationship of the WT and triple-mutant BK_Ca channels. The conductance was obtained from the peak tail currents. The currents of each channel were normalized by the maximum conductance of each channel for the vehicle-treated condition. **(D)** Negative shift of the $V_{1/2}$ value for WT and Ala mutant BK_Ca channels. To obtain $V_{1/2}$, each G-V curve was fitted using the Boltzmann function, $G/G_{max} = 1/[1 + \exp\{-zF(V - V_{1/2})/RT\}]$, where $G$ is the conductance. Data are presented as the mean ± SEM. **$P < 0.01$, ***$P < 0.001$ (two-sample $t$ test, n = 3–6).

channels close much faster in the presence of CTIBD than do WT channels.

### Single-channel recording of WT and triple-mutant BK_Ca channels with CTIBD treatment

We then determined the effects of CTIBD on WT and triple-mutant BK_Ca channels at the single-channel level. To identify the number of BK_Ca channels in the membrane patch, the current was recorded at or higher than 150 mV before collecting data. At 40 and 60 mV, 3 $\mu$M CTIBD increased the number of channel opening events in WT channels (Fig 6A). For the triple-mutant channel, however, 3 $\mu$M CTIBD yielded much fewer opening events than those in the WT channel (Fig 6B). The single-channel conductance of the WT and triple-mutant BK_Ca channels was not affected by 3 $\mu$M CTIBD (Fig 6C). Although the single-channel conductance of the WT BK_Ca channel was estimated as 293.3 ± 3.7 pS with the vehicle and 297.6 ± 1.8 pS with CTIBD, that of the triple mutant was 297.5 ± 4.3 pS with the vehicle and 303.6 ± 2.6 pS with CTIBD. The single-channel conductance was measured within the range of 20–60 mV and at a CTIBD concentration of 3 $\mu$M. Above 60 mV or at concentrations exceeding 3 $\mu$M of CTIBD, there remains the possibility that CTIBD can alter the single-channel conductance and $G_{max}$ (Figs 4C and S5).

CTIBD increased the open probability ($P_o$) for both WT and triple-mutant BK_Ca channels, but more so for the WT channel (Fig 6D). The $P_o$-values of the WT channel increased by 85-, 80-, 72-, 37-, and 18-fold at 20, 30, 40, 50, and 60 mV, respectively. In contrast, the $P_o$-values of the triple-mutant channel increased by only 9-, 14-, 13-, 10-, and 16-fold at 20, 30, 40, 50, and 60 mV, respectively. We also found that the difference in the CTIBD-induced $P_o$ increase in the WT and triple-mutant channels was voltage-dependent. Between 20 and 40 mV, the CTIBD-induced increase in the $P_o$ of the WT channel was more than fivefold that of the triple-mutant BK_Ca channel. At 50 and 60 mV, however, there was less difference in the fold increase between the WT and triple-mutant channels, presumably because the $P_o$ of the WT channel was nearly saturated.

### Association and dissociation kinetics of CTIBD binding to WT and triple-mutant BK_Ca channels

Next, we examined the association and dissociation kinetics of CTIBD in WT and triple-mutant BK_Ca channels. The macroscopic currents of WT and triple-mutant BK_Ca channels expressed in *Xenopus* oocytes were measured in an outside-out configuration in the presence of 3 $\mu$M intracellular $Ca^{2+}$. The channels were activated once every second using 150-mV voltage pulses of 50 ms from

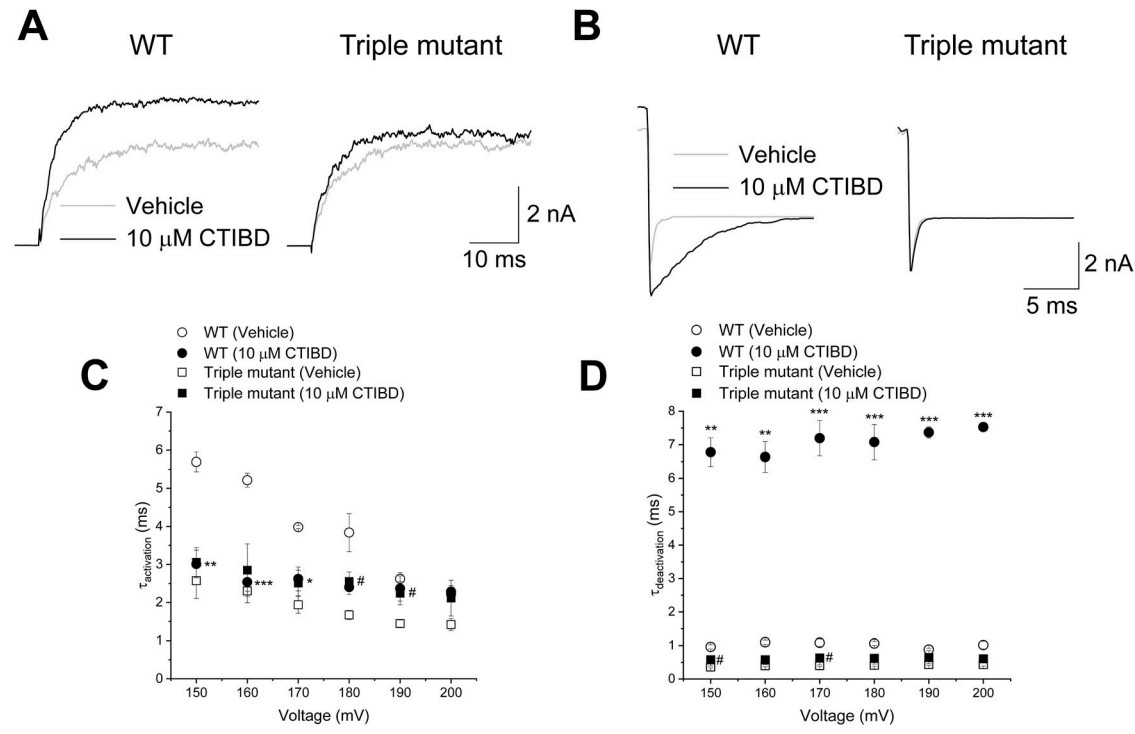

**Figure 5.   Effects of CTIBD on the activation and deactivation of the WT and triple-mutant (W22A/W203A/F266A) BK$_{Ca}$ channels.**
**(A, B)** Representative current traces of activation (A) and deactivation (B) are shown for the WT and triple-mutant BK$_{Ca}$ channels. The intracellular Ca$^{2+}$ concentration was 3 μM, and the vehicle was 0.1% DMSO. 10 μM of CTIBD was applied to the extracellular side of the channel. All current traces were obtained at 150 mV. **(C, D)** Activation time constant ($\tau_{activation}$) (C) and deactivation time constant ($\tau_{deactivation}$) (D) of the WT and triple-mutant BK$_{Ca}$ channels. Time constant values were obtained by fitting with a standard exponential function, y(t) = A$_1$exp(−t/$\tau_1$) + C, using the Clampfit program. All current traces were fitted individually. Data are presented as the mean ± SEM. *$P <$ 0.05, **$P <$ 0.01, ***$P <$ 0.001 (two-sample $t$ test, n = 3, compared with the vehicle-treated condition of each WT BK$_{Ca}$ channel). #$P <$ 0.05 (two-sample $t$ test, n = 4, compared with the vehicle-treated condition of each triple-mutant BK$_{Ca}$ channel).

a −100-mV holding voltage. When 10 μM CTIBD was applied, the WT channel current reached the maximum level slightly faster than did the triple mutant (Fig 7A). When CTIBD was washed out, however, the WT channel current was decreased much more slowly than the mutant channel current (Fig 7B). Based on the standard exponential fitting of current traces, the time constant of the WT channel was estimated as 3.1 ± 0.1 s for association and 51.3 ± 1.7 s for dissociation. However, the time constants for the triple mutant were 4.7 ± 0.5 s for association and 27.8 ± 0.8 s for dissociation (Fig 7C and D). These results indicate that CTIBD associates slowly with and dissociates much more rapidly from the triple-mutant channel compared with its actions on the WT BK$_{Ca}$ channel, meaning that the binding affinity of CTIBD is decreased when three residues, W22, W203, and F266, of the BK$_{Ca}$ channel are simultaneously mutated to Ala. This interpretation is supported by a molecular docking study of CTIBD. In the WT channel, the best docking pose reproduces the experimental binding orientation, with a root mean square deviation of 1.2 Å (Fig S6A). However, the triple mutant reveals an enlarged binding site, enabling altered interactions of CTIBD1 in a different orientation (Fig S6B). Notably, in the orientation most similar to the experimental structure, both the grid score (−50 kcal/mol) and van der Waals energy (−10 kcal/mol) were higher (less negative) than those of the WT channel–CTIBD complex (−56 and −13 kcal/mol, respectively), suggesting the weak binding affinity and structural instability of the triple mutant–CTIBD1 complex.

## Discussion

In this study, we investigated the mechanism of a newly discovered BK$_{Ca}$ channel activator, CTIBD, and identified its binding sites. BK$_{Ca}$ channel activation is controlled by intrinsic pore gating, Ca$^{2+}$ sensing, voltage sensing, and their coupling (Horrigan & Aldrich, 2002). To identify the component(s) responsible for CTIBD channel activation, we determined the role of Ca$^{2+}$ sensing in CTIBD-mediated channel activation. We found that CTIBD activated a truncated BK$_{Ca}$ channel (rSlo1-Kv-minT) lacking the intracellular Ca$^{2+}$-sensing domain (Fig 1), indicating that CTIBD activation is not mediated through a Ca$^{2+}$-sensing mechanism of the channel and suggesting that CTIBD may not bind to the cytosolic Ca$^{2+}$-sensing domain at the C-terminus of the channel. Next, we determined whether CTIBD influences the voltage-sensing mechanism of the BK$_{Ca}$ channel for activation. At highly negative voltages for which the voltage sensors are mainly in a resting, deactivated state, we observed that single rSlo1 channels were activated robustly by CTIBD (Fig 2). Although the $V_{1/2}$, which represents voltage dependence of the channel activation, is altered by CTIBD (Fig 4), the open probability within −90 to −60 mV (Fig 2) and the cryo-EM structure (Fig 3) make it clear that the compound does not physically interact with the voltage sensor. These results indicate that CTIBD binding does not affect the voltage-sensing mechanism or the coupling of voltage sensing to pore opening of the channel.

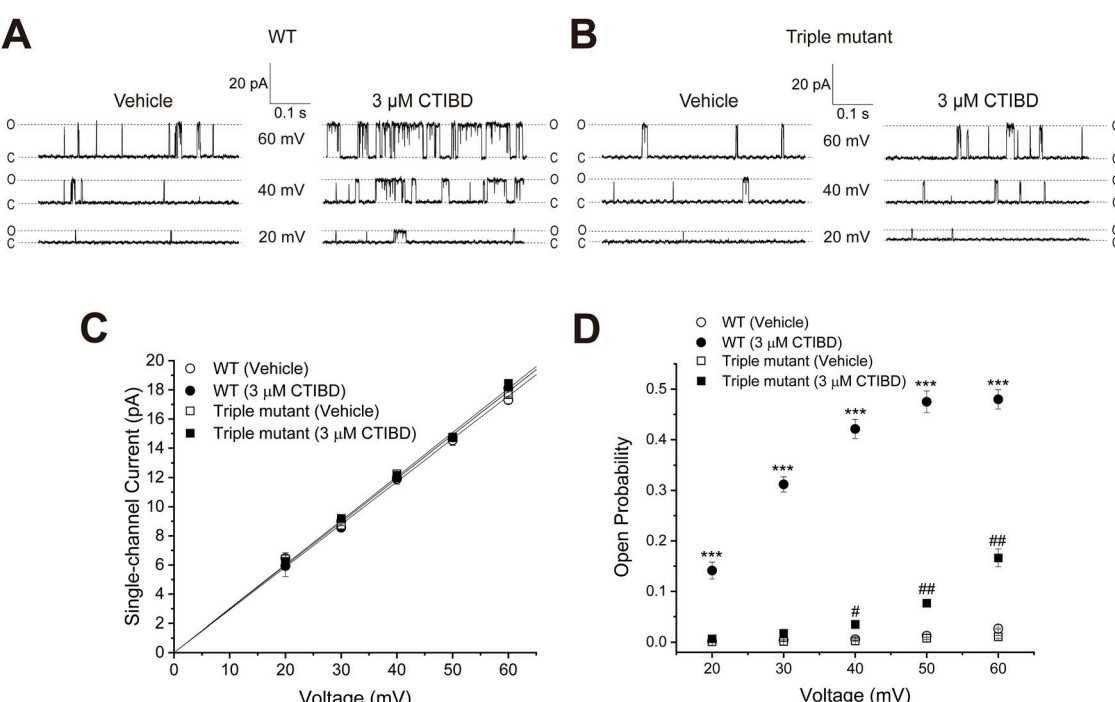

**Figure 6. Effects of CTIBD on WT and triple-mutant (W22A/W203A/F266A) BK$_{Ca}$ channels at the single-channel level.**
**(A, B)** Representative single-channel current recordings for WT (A) and triple-mutant (B) BK$_{Ca}$ channels at 20, 40, and 60 mV are shown. The line at the bottom represents the closed state, and the line at the top represents the open state of the channel. The intracellular Ca$^{2+}$ concentration was 3 $\mu$M, and the vehicle was 0.1% DMSO. The extracellular side of the channel was treated with 3 $\mu$M of CTIBD. **(C)** Single-channel current–voltage relationship of WT and triple-mutant BK$_{Ca}$ channels. The unitary current amplitude of the channel was obtained with 3 $\mu$M intracellular Ca$^{2+}$. Data points on the graphs were obtained by all-points histogram fitting using a Gaussian function. The single-channel conductance was estimated from the slope by fitting with a linear function. **(D)** $P_O$ of WT and triple-mutant BK$_{Ca}$ channels. The $P_O$ was obtained by dividing the N$P_O$ by N. N is the number of channels in the membrane patch, and N$P_O$ is the total open-state probability of N channels. The number of channels on the patch membrane was calculated by dividing the maximum current level at ≥150 mV by the single-channel conductance. Data are presented as the mean ± SEM. ***$P$ < 0.001 (two-sample $t$ test, n = 3, compared with the vehicle-treated condition of each WT BK$_{Ca}$ channel). #$P$ < 0.05, ##$P$ < 0.01 (two-sample $t$ test, n = 3, compared with the vehicle-treated condition of each triple-mutant BK$_{Ca}$ channel).

To understand the binding mode of CTIBD at the molecular level, we determined the cryo-EM structure of the BK$_{Ca}$ channel in complex with CTIBD. The presence of two CTIBD molecules on each subunit of the BK$_{Ca}$ channel provides a valuable mechanistic understanding of the structure–activity relationship. In our previous study (Lee et al, 2021), we observed that CTIBD activates the BK$_{Ca}$ channel in two distinct phases. In the presence of CTIBD, the channel current showed a rapid initial increase within seconds, followed by a slow and gradual enhancement over several minutes. These findings suggest that the BK$_{Ca}$ channel harbors at least two binding sites for CTIBD, each with distinct affinity and accessibility to the compound. Consistent with these electrophysiological data, our structure reveals that two molecules of CTIBD bind to the channel, one in the hydrophobic cavity and the other in a region near the turret loop (Fig 3A and B). Specifically, despite CTIBD2 being fully exposed to the membrane face, its secure binding to the channel occurred through π-stacking with W263 of the turret loop and van der Waals interactions with CHS (Fig 3C). This finding was unexpected because we anticipated that the CTIBD molecules would partially or entirely insert into a hydrophobic pocket formed by the transmembrane helices in a classic "key in a lock" manner, similar to the observed binding mode of CTIBD1. One possible explanation for this unexpected result is that the CTIBD2 binding

site may not function as a binding site under physiological conditions. The CTIBD2 binding site is observed under conditions treated with high concentrations of CTIBD using cryo-EM and found to be stabilized in the presence of CHS. In addition, the effects of CTIBD are almost negligible in the triple-mutant channel, further supporting the possibility that the occupancy of CTIBD to this second site is artifactual under experimental conditions or not physiologically functional at least. In line with the functional data, our structure reveals that the CTIBD molecules occupy the extracellular side of the pore-forming transmembrane helices to activate the channel (Fig S4), rather than targeting specific regions involved in modulating cytosolic Ca$^{2+}$ binding or membrane voltage (Fig 3A and B). By targeting this upper part of the transmembrane helices, therefore, future rational drugs can be designed to modulate BK$_{Ca}$ channel activity in a more selective and precise manner.

Further investigations involved mutagenesis and electrophysiology of the most important CTIBD-interacting residues (W22, W203, and F266). Although most mutants exhibited significantly reduced activation by CTIBD, the triple mutant (W22A/W203A/F266A) displayed the smallest channel currents and narrowest $G$-$V$ shift compared with those of the WT channel (Fig 4). All experiments in this study were conducted under 3 $\mu$M or 10 $\mu$M Ca$^{2+}$ concentrations. Therefore, it is possible that the efficacy of CTIBD could be influenced by changes in

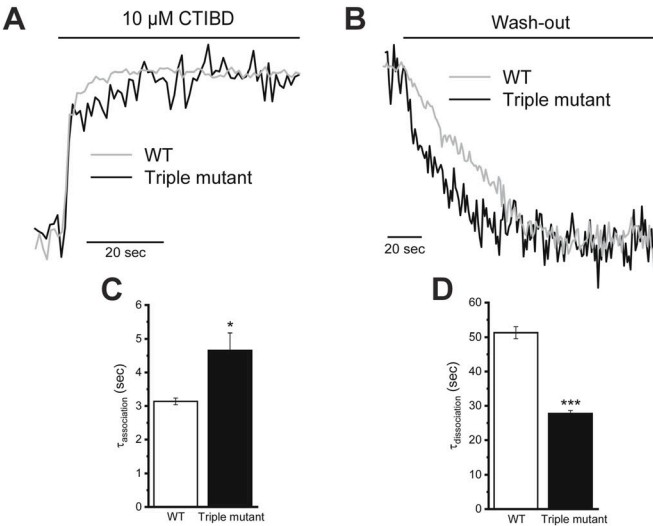

**Figure 7. Association and dissociation kinetics of CTIBD binding to WT and triple-mutant (W22A/W203A/F266A) BK_Ca channels.**

**(A, B)** Representative traces of the association (A) and dissociation (B) of CTIBD with the WT (gray) and triple-mutant (black) BK_Ca channels. Each trace was normalized and superimposed with the same time course. BK_Ca channels were activated once every second by a 100-mV pulse. The duration of each voltage pulse was 50 ms, and the holding voltage was −100 mV. Macroscopic BK_Ca channel currents were recorded once every second. Data points were obtained from the peak tail currents, and each data point was connected with a solid line. **(C, D)** Association time constant ($\tau_{association}$) (C) and dissociation time constant ($\tau_{dissociation}$) (D) values of WT and triple-mutant BK_Ca channels. Time constant values were obtained by fitting with a standard exponential function, $y(t) = A_1 \exp(-t/\tau_1) + C$, using the Origin 2021 program. All traces were fitted individually. Data are presented as the mean ± SEM. *$P < 0.05$, ***$P < 0.001$ (two-sample $t$ test, n = 3).

the activation $V_{1/2}$ under different $Ca^{2+}$ concentrations. Even so, these results strongly indicate that these three residues are crucial, if not essential, for the binding and activation of CTIBD. It is noteworthy that CTIBD can activate the triple mutant weakly but significantly. The origin of this remaining activation effect of the drug remains to be determined. The complex structure of the triple mutant with CTIBD could provide further information on this aspect.

In summary, we identified the binding sites of CTIBD, a potent new activator of BK_Ca channels, elucidating its interaction mechanism at the molecular level. By occupying the hydrophobic cavity and displacing lipid molecules on the extracellular side of the transmembrane helices, CTIBD plays a pivotal role in stabilizing the channel in its activated state. These sites have never been identified as a potential target for channel modulation and thus are worth investigating in more detail in the future. The results of this study provide the molecular mechanism of CTIBD-induced activation and valuable insights into the development of new small molecules for the treatment of BK_Ca-mediated diseases.

# Materials and Methods

### Synthesis and preparation of CTIBD

The chemical synthesis of CTIBD was described previously (Lee et al, 2021). For the patch-clamp experiments, CTIBD was dissolved in DMSO (Sigma-Aldrich) to create a 100-mM stock solution that was stored at −20°C until it was used for the experiments described below.

### Cloning and protein expression and purification

The human BK_Ca protein was prepared as described previously (Tao & MacKinnon, 2019), with minor modifications. Briefly, a codon-optimized human $BK_{Ca}$ gene (residues 1–1,114; Gene Universal) was cloned between the XhoI and NotI sites in the modified pEG BacMam vector containing a C-terminal thrombin-cleavable eGFP and a decahistidine (10×His) tag (Goehring et al, 2014). Bacmids of BK_Ca were generated using the plasmids by transforming them into DH10Bac *Escherichia coli* cells. The bacmid-produced colonies were selected using 5-bromo-4-chloro-3-indolyl-$\beta$-D-galactopyranoside (X-Gal) for blue-white screening. The high-purity bacmid was then transfected into *Spodoptera frugiperda* (Sf9) cells using the Cell-fectin II transfection reagent (Gibco) to produce recombinant baculovirus. Baculovirus-infected HEK293S GnTi⁻ cells were cultured for 20 h at 37°C under 8% $CO_2$ using FreeStyle 293 expression medium (Gibco). After treatment with 10 mM sodium butyrate, the cells were further cultured for ~48 h at 30°C before harvesting.

Cells were harvested by centrifugation at 14,000$g$ for 20 min and resuspended in a buffer containing 20 mM Tris–HCl, pH 8.0, 320 mM KCl, 10 mM $CaCl_2$, 10 mM $MgCl_2$, 10 $\mu$g/ml DNaseI, and 0.1 mM phenylmethylsulfonyl fluoride (GoldBio). The cells were lysed by sonication using Branson Sonifier equipped with a 19-mm microtip at 40% amplitude, with three cycles of 10 s on and 10 s off. The cell membrane was collected by ultracentrifugation at 240,000$g$ for 1 h. The proteins were solubilized with 1% (wt/vol) n-dodecyl-$\beta$-D-maltopyranoside (Anatrace) and 0.1% (wt/vol) CHS (Anatrace) for 2 h with gentle shaking. Nonsolubilized materials were removed by centrifugation at 240,000$g$ for 30 min, and the protein was subsequently purified using anti-GFP DARPin-based affinity chromatography (Hansen et al, 2017). The BK_Ca protein bound to the resin was washed with 10 column volumes of wash buffer containing 20 mM Tris–HCl, pH 8.0, 450 mM KCl, 10 mM $CaCl_2$, 10 mM $MgCl_2$, 0.01% (wt/vol) glyco-diosgenin (Anatrace), and a lipid mixture of 0.05 mg/ml 1-palmitoyl-2-oleoyl-sn-glycero-3-phosphoethanolamine (POPE), POPC, and 1-palmitoyl-2-oleoyl-sn-glycero-3-phosphate (POPA) in a 5:5:1 (w:w:w) ratio. The BK_Ca protein was eluted from the resin by thrombin on-column cleavage. The eluted protein was further purified using gel filtration chromatography on a Superose 6 Increase 10/300 GL column (GE Healthcare) with a buffer containing 20 mM Tris–HCl, pH 8.0, 450 mM KCl, 10 mM $CaCl_2$, 10 mM $MgCl_2$, 0.01% (wt/vol) glyco-diosgenin, and a lipid mixture of 0.05 mg/ml POPE, POPC, and POPA in a 5:5:1 (w:w:w) ratio. The fractions containing the tetrameric BK_Ca protein were pooled and concentrated to 5.5 mg/ml (44 $\mu$M) for cryo-EM grid preparation. All purification steps were performed at 4°C or on ice.

### Cryo-EM grid preparation and data collection

To obtain the BK_Ca-CTIBD complex, we mixed the purified protein with 1 mM CTIBD and incubated it on ice for 30 min before grid preparation. A total of 3 $\mu$l of the sample was then applied to a freshly glow-discharged 300-mesh Au R1.2/1.3 holey carbon grid

(Quantifoil). The grids were blotted for 3.5 s at a blot force of 0 and plunge-frozen in liquid ethane using a Vitrobot Mark IV (Thermo Fisher Scientific) under 100% humidity at 4°C. A total of 4,806 datasets were collected on a 200-kV Glacios transmission cryo-electron microscope equipped with a Falcon 4i direct electron detector (Thermo Fisher Scientific). All data were collected automatically using EPU software (Thermo Fisher Scientific) using a pixel size of 1.41 Å/pixel (nominal magnification of 73,000×) and a defocus range of −1.4 to −2.6 $\mu$m. The total dose was 50 e$^-$/Å$^2$ with an exposure time of 13.67 s, at an electron dose rate of 7.27 e$^-$/pixel/s (~1.0 e$^-$/Å$^2$/frame).

## Cryo-EM data processing

A detailed processing workflow is presented in Fig S7A. Briefly, data processing was performed using CryoSPARC v3.3.2 (Punjani et al, 2017). The dataset was corrected for beam-induced motion and dose-weighting using MotionCor2 (Zheng et al, 2017). After the contrast transfer functions on the resulting micrographs were estimated, low-quality micrographs were excluded from downstream processing (Fig S8A). Initial particle templates were generated by first selecting 23,000 particles using Blob Picker and 2D classification (Fig S8B). Good 2D classes were used as templates to automatically select particles from all micrographs via Template Picker. After three rounds of 2D classification, the best 2D classes containing 576,770 particles were used to generate an ab initio model, which was then subjected to heterogeneous and homogeneous 3D refinement, followed by global/local contrast transfer function refinement, local motion correction, and nonuniform refinement (Fig S8C and D) (Punjani et al, 2020). The final map was estimated to have resolutions of 4.2 and 3.9 Å using C1 and C4 symmetry, respectively, based on the gold-standard FSC with a cutoff of 0.143 (Fig S7B) (Bell et al, 2016). The local resolution was calculated in the two half-maps using CryoSPARC. The final map underwent post-processing using ResolveCryoEM in the PHENIX suite to improve the quality of the map and the local resolution at the CTIBD binding site (Fig S8E) (Adams et al, 2010).

## Model building and refinement

A starting model for the BK$_{Ca}$-CTIBD complex was generated by docking an atomic model of the cryo-EM structure of the open human BK$_{Ca}$ channel (PDB ID 6V38) (Tao & MacKinnon, 2019) onto the EM density map using UCSF Chimera (Pettersen et al, 2004). The position and orientation of the coordinates were adjusted by rigid body refinement and morphing using the phenix.real_space_refine program from the PHENIX suite (Adams et al, 2010). The model was further refined by iterative rounds of manual model building using Coot (Emsley et al, 2010) and real-space refinement in PHENIX. The weak-density regions were built as polyalanine regions. The quality of the structure was validated using MolProbity (Chen et al, 2010), and cross-validation was performed to confirm that over-refinement did not occur. The molecular graphics were prepared using PyMOL (https://pymol.org/), UCSF Chimera, and Chimera X (http://www.rbvi.ucsf.edu/chimera/). The refinement and validation statistics of the BK$_{Ca}$-CTIBD complex are summarized in Table S4 and Supplemental Data 1.

## Molecular docking

UCSF Dock 6.10 was used to perform molecular docking (Allen et al, 2015). To avoid potential errors and accelerate the simulation, we performed docking using only the transmembrane regions of the adjacent two subunits, excluding the cytoplasmic RCK domain. The protein preparation for docking, such as deleting solvent molecules, adding hydrogen atoms, and assigning charges to protein atoms, was performed with UCSF Chimera (Pettersen et al, 2004) using the default settings of the Dock Prep tool. The binding site of CTIBD1 was determined by "sphgen" estimation with a minimum sphere radius of 1.4 Å. The grid files were generated using a grid box of dimensions 29 Å × 26 Å × 23 Å, with a default grid spacing of 0.3 Å. After the CTIBD's energy was minimized through minor conformational changes, a flexible docking procedure was followed by the calculation of 1,000 orientations. The poses were viewed using UCSF Chimera's ViewDock tool, and the best scoring poses were ranked based on the DOCK 6 grid score and van der Waals energy (vdw_energy).

## Functional expression of the BK$_{Ca}$ channel in *Xenopus* oocytes

The rat *Slo1* gene (r*Slo1*) was expressed in *Xenopus laevis* oocytes for electrophysiological recordings. The cloning and expression of r*Slo1* (cDNA sequence GenBank accession number, AF135265) were performed using an oocyte expression vector (pNBC1.0) as described previously (Ha et al, 2006). Oocytes were surgically extracted from the ovarian lobes of *X. laevis* at stages V–VI (Nasco). The oocytes were placed in Ca$^{2+}$-free oocyte Ringer's (OR) culture medium (86 mM NaCl, 1.5 mM KCl, 2 mM MgCl$_2$, and 10 mM Hepes, pH 7.6) containing 3 mg/ml collagenase for 1.5–2 h at RT to remove the follicular cell layer. The oocytes were rinsed with Ca$^{2+}$-free OR medium and then rinsed with ND-96 medium (96 mM NaCl, 2 mM KCl, 1.8 mM CaCl$_2$, 1 mM MgCl$_2$, 5 mM Hepes, and 50 g/ml gentamicin, pH 7.6). The rinsed oocytes were stabilized by incubating them in ND-96 medium at 18°C for at least 24 h. Approximately 50 ng (macroscopic recording) or 1 ng (single-channel–level recording) of synthesized r*Slo1* mRNA, prepared in 50 nl of nuclease-free water, was injected into each oocyte, which was then incubated at 18°C for 1–3 d in ND-96 medium. Before electrophysiological recordings, the vitelline membrane was removed using fine forceps.

## Electrophysiological recordings and data analysis

The giga-ohm seal patch-clamp method was used for both macroscopic and single-channel recordings of BK$_{Ca}$ channels in an outside-out configuration, as described previously (Ha et al, 2006). Pipettes made of borosilicate glass (WPI) were created by pulling and fire-polishing, and their resistance was 4–6 MΩ for macroscopic recordings. The patch pipettes used for single-channel recordings had a fire-polished resistance of 5–8 MΩ. During the recordings, the channel currents were amplified using an Axopatch 200B amplifier (Molecular Devices), and currents were low-pass–filtered at 10 kHz with a four-pole Bessel filter. Currents were digitized at a rate of 10 or 20 points/ms using a Digidata 1200A (Molecular Devices) to accurately record and analyze the data. To record the macroscopic currents of BK$_{Ca}$ channels, voltage-clamp pulses ranging from −80

to 180 mV were applied in increments of 10 mV while holding the resting potential at −100 mV. Standard single-channel patch-clamp recording protocols were used to record the single-channel currents of $BK_{Ca}$ channels. Both macroscopic and single-channel recordings were performed in an outside-out configuration. During experiments, oocytes were immersed in the recording solution (120 mM potassium gluconate, 10 mM Hepes, pH 7.2, 4 mM KCl, and 5 mM EGTA). The concentration of free $Ca^{2+}$ in the intracellular solution was determined using the MaxChelator program (Patton et al, 2004), which calculated the values of HEDTA, EGTA, and free $Ca^{2+}$. HEDTA was used to prepare calcium solutions greater than 1 $\mu$M, whereas EGTA was used to prepare calcium solutions of 1 $\mu$M or lower (Patton et al, 2004). The intracellular solution contained 116 mM KOH, 10 mM Hepes, and 4 mM KCl, pH 7. The solution exchange was completed within 0.5 s. Data acquisition and analysis were performed using Clampex 8.0, Clampfit 11.0.3, and Origin 2021 software (OriginLab Corporation). To reduce the noise, single-channel recording data were low-pass–filtered at 1 kHz post-acquisition with an eight-pole Bessel filter using Clampfit 11.0.3 software. Data are presented as the mean ± SEM, and a two-sample *t* test was used for statistical analysis.

### Mutagenesis of the ion channel

The r*Slo1* gene (GenBank, AF135265) has unique Cla I (ATCGAT) and Pac I (TTAATTAA) restriction sites located at amino acid positions 152 and 584. A BamH I (GGATCC) restriction site is located in the oocyte expression vector (pNBC1.0). To create mutations, we performed polymerase chain reactions with mutagenic primers using CloneAmp HiFi PCR Premix (Clontech). The resulting amplified DNA fragments were used to replace the WT r*Slo1* gene in the pNBC1.0 oocyte expression vector using the BamH I or Cla I and Pac I restriction sites. To generate the truncated mutant $BK_{Ca}$ channel (r*Slo1*-Kv-minT), the C-terminal domain of r*Slo1* was replaced with a sequence encoding a short segment containing 11 amino acid residues from the C-terminus of Kv1.4 (Budelli et al, 2013). This mutation was constructed using site-directed mutagenesis, as described previously (Lee et al, 2012). The protein sequence information was obtained from the PDB file, accession number 6V22, which represents the cryo-EM structure of the $Ca^{2+}$-bound h*Slo1*-$\beta$4 channel complex (Tao & MacKinnon, 2019). All mutations were verified by sequencing.

## Data Availability

The refined atomic coordinates and cryo-EM density map of the $BK_{Ca}$-CTIBD complex have been deposited in the PDB and the Electron Microscopy Data Bank under accession numbers 8Z3S and EMD-38753, respectively.

## Supplementary Information

## Acknowledgements

This work was supported by the National Research Foundation of Korea (NRF) grants (NRF-2019M3E5D6063908 and NRF-2022R1A2C1005532) and a postdoctoral fellowship (NRF-2021R1A6A3A01086747) funded by the Korean government (MSIT). This research was also supported by the Korea Drug Development Fund funded by the Ministry of Science and ICT, Ministry of Trade, Industry, and Energy, and Ministry of Health and Welfare (RS-2023-00258812) and by a "GIST Research Project" grant funded by the GIST in 2023. We would like to express our gratitude to Prof. Jaebong Kim of Hallym University for providing *X. laevis* from the Korean *Xenopus* Resource Center for Research (ROK).

### Author Contributions

N Lee: conceptualization, data curation, formal analysis, validation, investigation, visualization, methodology, project administration, and writing—original draft, review, and editing.
S Kim: conceptualization, data curation, formal analysis, validation, investigation, visualization, methodology, project administration, and writing—original draft, review, and editing.
NY Lee: data curation, formal analysis, validation, visualization, and writing—review and editing.
H Jo: resources.
P Jeong: conceptualization and resources.
HS Pagire: resources.
SH Pagire: resources.
JH Ahn: resources.
MS Jin: conceptualization, supervision, funding acquisition, validation, visualization, methodology, project administration, and writing—original draft, review, and editing.
C-S Park: conceptualization, supervision, funding acquisition, validation, visualization, methodology, project administration, and writing—original draft, review, and editing.

### Conflict of Interest Statement

N Lee, HS Pagire, SH Pagire, JH Ahn, and C-S Park have submitted a patent application (PCT/KR2021/013819) for this family of molecules.

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
