## [Reviewer comments · Life Science Alliance]

Life Science Alliance

Activation mechanism and novel binding sites of the BKCa channel activator CTIBD

Narasaem Lee, Subin Kim, Na Young Lee, Heeji Jo, Pyeonghwa Jeong, Haushabhau Shivaji Pagire, Suvarna Haushabhau Pagire, Jin Hee Ahn, Mi Sun Jin and Chul-Seung Park

DOI: <https://doi.org/10.26508/lsa.202402621>

Corresponding authors: Prof. Chul-Seung Park and Prof. Mi Sun Jin (Gwangju Institute of Science and Technology)

Review Timeline:

Submission Date:	2024-01-26
Editorial Decision:	2024-03-07
Revision Received:	2024-05-26
Editorial Decision:	2024-06-18
Revision Received:	2024-07-04
Editorial Decision:	2024-07-08
Revision Received:	2024-07-12
Accepted:	2024-07-15

Transaction Report:

March 7, 2024

Re: Life Science Alliance manuscript #LSA-2024-02621-T

Prof. Chul-Seung Park
Gwangju Institute of Science and Technology
Life Sciences
Dept. of Life Science Gwanju Institute of Science & Technology 1 Oryng-dong Buk-gu
Gwang-ju, Buk-Gu 500-712
Korea

Dear Dr. Park,

Thank you for submitting your manuscript entitled "Activation mechanism and novel binding site of the BKCa channel activator CTIBD" to Life Science Alliance. The manuscript was assessed by expert reviewers, whose comments are appended to this letter. We invite you to submit a revised manuscript addressing the Reviewer comments.

Thank you for this interesting contribution to Life Science Alliance. We are looking forward to receiving your revised manuscript.

Sincerely,

B. MANUSCRIPT ORGANIZATION AND FORMATTING:

Reviewer #1 (Comments to the Authors (Required)):

In a previous study, the authors reported that "CTIBD", an BKCa channel activator enhances current by shifting the voltage dependence of activation (by -85 mV at 10 μ M) and increasing the maximal conductance of channels. In the present study, a more detailed description of its molecular mechanism and site of action was investigated by cryo-EM (of homotetrameric human Slo1 channels complexed with CTIBD), site-directed mutagenesis of potential compound-interacting residues identified in the structure, and functional analysis (voltage clamp) of wild-type and mutant channels. Two Trp and one Phe residues located on the extracellular surface of the channel were found to be major determinants of CTIBD activity as their mutation to Ala residues greatly reduced the effect of the compound on current magnitude and shift of the G-V relationship. In addition, one CTIBD molecule interacts with an adjacent cholesteryl hemisuccinate. Additional experiments with truncated channels and choice of voltage for channel activation suggest that the Ca²⁺- and voltage-sensing mechanisms are not required for CTIBD-dependent channel activation. The findings should assist drug-design efforts to discovery further BKCa channel activating compounds that may find clinical utility in treatment of overactive bladder syndrome and possibly other ailments.

Although previously reported, it would be helpful if the first data of the Results section presented the effects of the compound on wild-type channels under normal physiological conditions, similar to what is now shown in Fig 4c. This is important for comparison to the data shown in Fig 1 for channels missing its Ca²⁺-sensing region where although CTIBD increases current magnitude, the effect is much less than that observed with channels comprised of fully intact alpha-subunits. In contrast to your statement that: "the Ca²⁺-sensing domain is not essential for CTIBD-mediated BKCa channel activation" it would appear that this domain is in fact required for the majority of the effect of the compound.

Although CTIBD clearly activates BKCa channels at voltages where the voltage sensors are normally in a resting conformation (Fig. 2) does not prove that the compound does not interact with or bind to the voltage sensors. In fact, the dramatic shift in the voltage dependence of activation would seem to suggest that voltage sensor interaction was a primary mechanism of action. Please elaborate on why you would conclude that these findings necessarily suggest that voltage sensing is not required for CTIBD activity. The cryo-EM structure and mutagenesis results do support the idea that the compound does not interact physically with the voltage sensor domain, but the electrophysiological results of the wild-type channel (Fig. 2) do not.

Was there any difference in the configuration of the ion conduction pathway between the CTIBD-bound and the apo BKCa tetramer structure that would suggest why conductance is increased in the bound state, or were no differences expected because both structures are in an open state? The triple mutant channel is clearly much less affected by CTIBD compared to wild-type channels. Do you have any insights into the relevant importance of the two identified binding sites other than the effects of single mutants? What about relative binding affinities or interaction (cooperativity?) between the two binding sites? The data of Fig 7 presumably reflect the binding and unbinding rates from both binding sites, but does not provide any insights into the relative binding affinities of the different sites.

Activation and deactivation time constants were obtained by fitting current traces recorded using unphysiological voltage transients (membrane potentials of 150 to 170 mV are never obtained in vivo). I realize such positive activating voltages and the return voltage of -100 mV were employed to maximize current magnitudes, but it would also be of interest to know if kinetics are also altered by this compound over a range of voltage experienced by smooth muscle cells in intact muscle.

Reviewer #2 (Comments to the Authors (Required)):

Lee and colleagues studied the mechanism of CTIBD modulation of BK channels. By recording CTIBD effects on a truncated BK lacking the gating ring and at negative voltages where the VSD was supposed at the resting state, the authors concluded that CTIBD modulates the channel by enhancing pore opening independently from Ca²⁺ or voltage dependent activation. The authors identified two CTIBD binding sites, CTIBD1 and CTIBD2, by solving the structure of the channel-CTIBD complex using cryo-EM. Mutations of the CTIBD-interacting residues reduced CTIBD modification of GV, kinetics and single channel open probability. These results in general support the mechanism and CTIBD binding. Some comments and questions are as follows.

- 1) Fig 1. The currents are really small (Fig 1B). Are these real BK channel currents or contaminated by endogenous currents? A blocking of the currents by BK specific inhibitors or at least from a sham injected oocyte should be shown as control.
- 2) Is the channel in Figure 2 WT BK or CTD-truncated BK? In many papers in which the P_o was measured at negative voltages, such as Horrigan et al JGP 2009, the P_o was between 10^{-7} and 10^{-6} . In this manuscript the P_o is above 10^{-6} (Fig 2E). Can the authors explain such a difference? The authors may want to show the data that were used for calculating the P_o , such as histograms to show the quality of the data.
- 3) Fig 3. The CTIBD2 site is stabilized by CHS. Is this binding real or an artifact? The triple mutations eliminated most of the CTIBD caused GV shift. Since all three residues in the triple mutation are important for the CTIBD1 site, the mutational results seem to suggest that the CTIBD2 site did not contribute to modulation of the channel.

Reviewer #3 (Comments to the Authors (Required)):

The authors present very interesting cryo-EM and electrophysiology data in their manuscript which demonstrates the likely binding site for the BK channel opener CTIBD. I do not have significant expertise in cryo-EM data so can't comment on the quality of this aspect of the work. However, the authors do present clear evidence for a binding site for CTIBD with the cryo-EM work and then use alanine mutations of these residues to examine if they reduce the efficacy of CTIBD. A triple mutant of W22A:W203A:F266A reduced the ability of CTIBD to shift activation $V_{1/2}$ by ~90%.

Major Points

1. I have a concern that the recorded currents were not maximally activated and G_{Max} was not accurately determined. Therefore the shifts in $V_{1/2}$ quoted in the manuscript may be problematic.
2. How can the authors be sure that the mutations didn't simply alter gating, rather than affect binding of the CTIBD? For example, is the Ca^{2+} sensitivity of each of the alanine mutants altered? Does 100 nM, 1 μ M and 10 μ M Ca^{2+} shift the $V_{1/2}$ of the mutants by a similar amount compared to the WT channel? If not, it's very difficult to be sure that the effect of the mutation was solely due to an effect on binding. Using inside out recordings may make these experiments easier and I presume that the CTIBD could be included in the patch pipette solution.
3. It would be really useful if the authors could also state the activation $V_{1/2}$ values for each of the mutants before and during CTIBD in a new table.
4. The authors mention that the currents were filtered at 1 KHz to reduce noise. Was the filtering carried out after data analysis and only used for presentation purposes?
5. Is there any difference in the effects of the CTIBD when GV curves are constructed from the steady state currents rather than the tail currents?
6. What effect does 10 μ M CTIBD have on the single channel recordings? Shown in Figure 6?
7. Figure EV7 is very difficult to read. It might be an idea to make a new multi-panel figure in which the effect of each mutation can be clearly seen. At present it is very difficult to make out the effects of most mutations.

Minor Points

1. In Abstract, please rephrase the penultimate sentence to clearly state that CTIBD was less effective in the triple mutant. It currently suggests that CTIBD actually reduced the open probability of these channels, which is wrong.
2. In the second sentence of the Introduction, insert "at physiological K^+ gradients and membrane potentials" after "through the channel"
3. Please insert "not" into the sentence about the truncated BK channel (Bottom of 1st paragraph of Intro) so that it reads "the large intracellular domain are *NOT* activated by an increase"
4. Introduction Paragraph 3, Last line and Paragraph 4, 6 lines from bottom. Please include references mentioning Cym04, BC5, omega-3 docosahexaenoic acid and the GoSlo family of BK channel openers since the molecular mechanisms of their mode of action is relevant.
5. Insert references in intro to the recently discovered LINGO family of BK regulatory subunits.

Point-by-point response to Reviewers

Reviewer #1 (Comments to the Authors (Required)):

In a previous study, the authors reported that "CTIBD", an BKCa channel activator enhances current by shifting the voltage dependence of activation (by -85 mV at 10 μ M) and increasing the maximal conductance of channels. In the present study, a more detailed description of its molecular mechanism and site of action was investigated by cryo-EM (of homotetrameric human Slo1 channels complexed with CTIBD), site-directed mutagenesis of potential compound-interacting residues identified in the structure, and functional analysis (voltage clamp) of wild-type and mutant channels. Two Trp and one Phe residues located on the extracellular surface of the channel were found to be major determinants of CTIBD activity as their mutation to Ala residues greatly reduced the effect of the compound on current magnitude and shift of the G-V relationship. In addition, one CTIBD molecule interacts with an adjacent cholesteryl hemisuccinate. Additional experiments with truncated channels and choice of voltage for channel activation suggest that the Ca²⁺- and voltage-sensing mechanisms are not required for CTIBD-dependent channel activation. The findings should assist drug-design efforts to discovery further BK_{Ca} channel activating compounds that may find clinical utility in treatment of overactive bladder syndrome and possibly other ailments.

1. Although previously reported, it would be helpful if the first data of the Results section presented the effects of the compound on wild-type channels under normal physiological conditions, similar to what is now shown in Fig 4c. This is important for comparison to the data shown in Fig 1 for channels missing its Ca²⁺-sensing region where although CTIBD increases current magnitude, the effect is much less than that observed with channels comprised of fully intact alpha-subunits. In contrast to your statement that: "the Ca²⁺-sensing domain is not essential for CTIBD-mediated BK_{Ca} channel activation" it would appear that this domain is in fact required for the majority of the effect of the compound.

⇒ If the calcium sensing domain is removed, the channel cannot be activated by increasing the calcium concentration, and therefore must be activated solely by increasing the membrane voltage. As a result, when comparing the G-V curve of the wild-type channel in Fig. 4 with the G-V curve of the mutant channel with the calcium sensing domain removed in Fig. 1, the G-V curve in Fig. 1 shifts much further to the right than the G-V curve in Fig. 4. Consequently, the current in Fig. 1 appears lower compared to Fig. 4, making it seem as if the activation effect of CTIBD is diminished. In other words, the G-V curve in the range of -60 to -30mV in Fig. 4, where the channel begins to activate, appears similar to the G-V curve in the range of 210 to 250mV in Fig. 1. We planned an experiment to see if we could achieve a G-V curve similar to that of Fig. 4 by sufficiently increasing the membrane voltage to saturate the current in the channel without the calcium sensing domain. However, we regret that we could not proceed with further experiments because applying high membrane voltage caused the patch membrane to rupture.

Secondly, if the calcium sensing domain were essential for the channel activation effect of CTIBD, there would be no difference between the CTIBD-treated and vehicle-treated cases in the channel with the calcium sensing domain removed, as shown in Fig. 1. However, as seen in Fig. 1, CTIBD significantly increases the channel current even in the absence of the calcium sensing domain. Therefore, the assertion that “the Ca²⁺-sensing domain is not essential for CTIBD-mediated BK_{Ca} channel activation” is considered a reasonable argument.

2. Although CTIBD clearly activates BK_{Ca} channels at voltages where the voltage sensors are normally in a resting conformation (Fig. 2) does not prove that the compound does not interact with or bind to the voltage sensors. In fact, the dramatic shift in the voltage dependence of activation would seem to suggest that voltage sensor interaction was a primary mechanism of action. Please elaborate on why you would conclude that these findings necessarily suggest that voltage sensing is not required for CTIBD activity. The cryo-EM structure and mutagenesis results do support the idea that the compound does not interact physically with the voltage sensor domain, but the electrophysiological results of the wild-type channel (Fig. 2) do not.

⇒ Even at membrane voltage levels lower than the resting membrane potential (-90 to -60mV), CTIBD significantly increases the channel's open probability. This indicates that CTIBD can activate the channel even when the voltage-sensing function of the channel is not activated. Furthermore, if the voltage-sensing function affects CTIBD's channel activation effect, the slope of the G-V curve should change depending on whether CTIBD is present or not. However, comparing the slopes of the G-V curves between when CTIBD is applied and when it is not applied to wild-type BK_{Ca} channels (Fig. 4), no significant difference was observed. This means that the channel activation effect by CTIBD is not significantly influenced by the channel's voltage-sensing function.

Table. The G-V curve slopes of the WT BK_{Ca} channel under vehicle or 10 μM CTIBD-treated conditions were measured. To obtain the slope, each G-V curve was fitted using the Boltzmann function, $G/G_{max} = 1/[1 + \exp \{-zF(V - V_{1/2})/RT\}]$. Slope equals RT/zF.

	Vehicle (n = 5)	10 μM CTIBD (n = 5)	P-value (Two-sample t-test)
Slope of G-V curve	15.79 ± 1.66	18.36 ± 0.91	0.213 (Not significant)

3. Was there any difference in the configuration of the ion conduction pathway between the CTIBD-bound and the apo BK_{Ca} tetramer structure that would suggest why conductance is increased in the bound state, or were no differences expected because both structures are in an open state? The triple mutant channel is clearly much less affected by CTIBD compared to wild-type channels.

⇒ In our study, we observed that the CTIBD-bound BK_{Ca} channel adopts a conformation highly similar

to that of the open state. For further details, we have included a comparison figure of the ion conduction pathways of BK_{Ca} channel in the closed, open, and CTIBD-bound states. Please refer to page 5 and Figure S3.

4. Do you have any insights into the relevant importance of the two identified binding sites other than the effects of single mutants? What about relative binding affinities or interaction (cooperativity?) between the two binding sites? The data of Fig 7 presumably reflect the binding and unbinding rates from both binding sites, but does not provide any insights into the relative binding affinities of the different sites.

⇒ We agree that it is important to experimentally determine the relative binding affinities and cooperativity between the two binding sites. However, it is unlikely to achieve accurate measurements because the binding sites are very close, which is expected to hinder the ability to separately evaluate the characteristics of each site.

⇒ Alternatively, we performed a molecular docking study of CTIBD1 with the wild-type BK_{Ca} channel and its triple mutant. The results indicate that the mutations lead to weaker interactions with CTIBD1, primarily due to the loss of favorable interactions between the compound and surrounding residues. We also performed docking analysis of CTIBD2; however, it proved unsuccessful, probably due to the absence of CHS involvement during the simulation. For further details, refer to page 7 and Figure S6.

5. Activation and deactivation time constants were obtained by fitting current traces recorded using unphysiological voltage transients (membrane potentials of 150 to 170 mV are never obtained in vivo). I realize such positive activating voltages and the return voltage of -100 mV were employed to maximize current magnitudes, but it would also be of interest to know if kinetics are also altered by this compound over a range of voltage experienced by smooth muscle cells in intact muscle.

⇒ Reviewer #1's opinion that obtaining experimental data under physiological conditions would be much more interesting and meaningful is something I fully agree with. However, at low voltage levels under physiological conditions, it was not possible to accurately determine the time constant due to the low current levels resulting in large error values when the channel opens and closes, making proper fitting impossible. In future research, we will explore methods to elucidate the kinetics of channel activation and inactivation under physiological voltage levels.

Reviewer #2 (Comments to the Authors (Required)):

Lee and colleagues studied the mechanism of CTIBD modulation of BK channels. By recording CTIBD effects on a truncated BK lacking the gating ring and at negative voltages where the VSD was supposed

at the resting state, the authors concluded that CTIBD modulates the channel by enhancing pore opening independently from Ca^{2+} or voltage dependent activation. The authors identified two CTIBD binding sites, CTIBD1 and CTIBD2, by solving the structure of the channel- CTIBD complex using cryo-EM. Mutations of the CTIBD-interacting residues reduced CTIBD modification of GV, kinetics and single channel open probability. These results in general support the mechanism and CTIBD binding. Some comments and questions are as follows.

1. Fig 1. The currents are really small (Fig 1B). Are these real BK channel currents or contaminated by endogenous currents? A blocking of the currents by BK specific inhibitors or at least from a sham injected oocyte should be shown as control.

⇒ From the figure below, it can be seen that when only DEPC water without channel mRNA is injected into *Xenopus* oocytes, no current was observed except for an extremely small leak current. In contrast, when DEPC water containing *rSlo1-Kv-minT* mRNA is injected into oocytes, a larger current is observed in the same voltage range. This indicates that the current in Fig. 1 is from the *rSlo1-Kv-minT* channel expressed by *rSlo1-Kv-minT* mRNA.

(A) Representative current traces were obtained from oocytes' patch membranes under DEPC water or *rSlo1-Kv-minT* mRNA injected conditions. The intracellular Ca^{2+} concentration was 0 μM . Currents were recorded for each of the voltage pulses, which increased from -30 to $+250$ mV in 20 mV increments. The holding voltage was -100 mV. Voltage pulses were applied to the patch membrane for 100-ms under DEPC water injected conditions and for 50-ms under *rSlo1-Kv-minT* mRNA injected conditions.

(B) Current-voltage relationship of DEPC water and *rSlo1-Kv-minT* mRNA injected conditions. After voltage pulse initiation, the average current was obtained from the outward current between 35 and 45 ms. Data are presented as the mean \pm SEM.

2) Is the channel in Figure 2 WT BK or CTD-truncated BK? In many papers in which the P_o was

measured at negative voltages, such as Horrigan et al JGP 2009, the P_o was between 10^{-7} and 10^{-6} . In this manuscript the P_o is above 10^{-6} (Fig 2E). Can the authors explain such a difference? The authors may want to show the data that were used for calculating the P_o , such as histograms to show the quality of the data.

⇒ Most of the papers authored by Frank T. Horrigan used mSlo, the mouse BK_{Ca} channel, not rSlo (Horrigan FT, 2012, Horrigan FT & Aldrich RW, 2002, Horrigan FT & Ma Z, 2008). Therefore, it is natural that our paper, which used the rat BK_{Ca} channel, would yield different open probability values. The paper titled 'Molecular mechanisms underlying the effect of the novel BK channel opener GoSlo: Involvement of the S4/S5 linker and the S6 segment,' published in PNAS in 2015, used rat BK_{Ca} channels, and similar open probability values in the range of 10^{-6} to 10^{-5} , as observed in our data at -90 to -60mV, were confirmed (Webb TI et al, 2015).

3) Fig 3. The CTIBD2 site is stabilized by CHS. Is this binding real or an artifact? The triple mutations eliminated most of the CTIBD caused GV shift. Since all three residues in the triple mutation are important for the CTIBD1 site, the mutational results seem to suggest that the CTIBD2 site did not contribute to modulation of the channel.

⇒ In our previous study (Lee N et al, 2021), we observed that CTIBD activates the BK_{Ca} channel in two distinct phases. In the presence of CTIBD, the channel current showed a rapid initial increase within seconds, followed by a slow and gradual enhancement over several minutes. These findings suggest that the BK_{Ca} channel possesses at least two binding sites for CTIBD, each with differing affinity and accessibility to the compound. Hence, we do not consider CTIBD2 as nonspecific or artifact.

Reviewer #3 (Comments to the Authors (Required)):

The authors present very interesting cryo-EM and electrophysiology data in their manuscript which demonstrates the likely binding site for the BK channel opener CTIBD. I do not have significant expertise in cryo-EM data so can't comment on the quality of this aspect of the work. However, the authors do present clear evidence for a binding site for CTIBD with the cryo-EM work and then use alanine mutations of these residues to examine if they reduce the efficacy of CTIBD. A triple mutant of W22A:W203A:F266A reduced the ability of CTIBD to shift activation $V_{1/2}$ by ~90%.

Major Points

1. I have a concern that the recorded currents were not maximally activated and G_{Max} was not accurately determined. Therefore, the shifts in $V_{1/2}$ quoted in the manuscript may be problematic.

⇒ Our recorded data extends up to 200mV. As seen in the graph below, we conducted recordings over a sufficient voltage range up to 200mV, within which the current saturates adequately. However, at the higher voltage levels of 190 and 200mV, channel blocking by magnesium or calcium ions may occur. Therefore, we included data only up to -80 to 180mV in this paper. By recording up to 200mV, we obtained an accurate G_{max} value, so there should be no error in the $V_{1/2}$ value.

(A) Conductance-voltage relationship of the WT BK_{Ca} channel under vehicle or 10 μ M CTIBD treated conditions. The intracellular Ca²⁺ concentration was 3 μ M. The vehicle was 0.1% DMSO. CTIBD was applied to the extracellular side of the channel. The duration of the voltage pulses was 100 ms. Currents were recorded for each of the voltage pulses, which increased from -80 to 200 mV in 10-mV increments. The holding voltage was -100 mV. The conductance was obtained from the peak tail currents. The currents of each channel were normalized by the maximum conductance of each channel for the vehicle-treated condition. Data are presented as the mean \pm SEM.

2. How can the authors be sure that the mutations didn't simply alter gating, rather than affect binding of the CTIBD? For example, is the Ca²⁺ sensitivity of each of the alanine mutants altered? Does 100 nM, 1 μ M and 10 μ M Ca²⁺ shift the $V_{1/2}$ of the mutants by a similar amount compared to the WT channel? If not, it's very difficult to be sure that the effect of the mutation was solely due to an effect on binding. Using inside out recordings may make these experiments easier and I presume that the CTIBD could be included in the patch pipette solution.

⇒ To address the second question, recordings were conducted for the W22A/W203A/F266A triple mutant and wild-type BK_{Ca} channels at calcium concentrations of 1, 3 and 10 μ M. As shown in the figure and table below, the BK channel activation effects due to increasing calcium concentration were similar for both the triple mutant and wild-type channels, with no significant difference observed between them. This indicates that the triple mutant channel exhibits a level of activation increase similar to that of the wild type, not by CTIBD but by another activator, calcium ions. Therefore, the reduction in activation effect by CTIBD in the triple mutant channel is due to a decrease in the binding affinity of CTIBD rather than a change in the gating of the channel itself caused by the triple mutation.

(A) Conductance-voltage relationship of the WT and triple-mutant BK_{Ca} channels. The duration of the voltage pulses was 100 ms. Currents were recorded for each of the voltage pulses, which increased from -80 to 200 mV in 10 -mV increments. The holding voltage was -100 mV. The conductance was obtained from the peak tail currents. The conductance of each channel were normalized by the maximum conductance of each channel. Data are presented as the mean \pm SEM (two-sample t-test, $n = 3-5$).

(B) Negative shift of the $V_{1/2}$ value between 1μ M and 10μ M of intracellular Ca^{2+} concentration for WT and triple-mutant BK_{Ca} channels. To obtain $V_{1/2}$, each $G-V$ curve was fitted using the Boltzmann function, $G/G_{max} = 1/[1 + \exp \{-zF(V - V_{1/2})/RT\}]$, where G is the conductance. Data are presented as the mean \pm SEM. 'ns' means not significant (two-sample t-test, $n = 3-5$).

Table. $V_{1/2}$ value of 1 , 3 and 10μ M of intracellular Ca^{2+} concentration for WT and triple-mutant BK_{Ca} channels.

($n=3-5$)

$V_{1/2}$ (mV)	1μ M Ca^{2+}	3μ M Ca^{2+}	10μ M Ca^{2+}
WT	127.30 ± 5.36	103.09 ± 7.41	69.73 ± 1.57
Triple mutant(WWF)	162.52 ± 0.55	128.27 ± 1.66	106.84 ± 1.01

3. It would be really useful if the authors could also state the activation $V_{1/2}$ values for each of the mutants before and during CTIBD in a new table.

\Rightarrow We have added a table for the $V_{1/2}$ values of each mutant channel to the supplementary materials.

4. The authors mention that the currents were filtered at 1 KHz to reduce noise. Was the filtering carried out after data analysis and only used for presentation purposes?

⇒ For single-channel data analysis, we analyzed the original data after filtering it at 1 kHz. This was done to reduce noise and obtain more accurate open probability values.

5. Is there any difference in the effects of the CTIBD when GV curves are constructed from the steady state currents rather than the tail currents?

⇒ Steady-state current (outward current) show a phenomenon known as slow flattening. It means that the current shape spikes up and then decreases due to clamping and equipment specifications. Therefore, using tail current provides a more accurate value.

6. What effect does 10uM CTIBD have on the single channel recordings? Shown in Figure 6?

⇒ When treating with 10 μ M CTIBD, there is a phenomenon of patch membrane rupture in single-channel recordings at positive membrane voltages. In single-channel recordings, unlike macroscopic recordings, we need to continuously apply the desired membrane voltage (in this paper, 20~60 mV). While macroscopic recordings have a holding voltage where the patch membrane can rest, single-channel recordings do not. Consequently, the membrane is subjected to a lot of damage, and when treated with 10 μ M CTIBD, the damage from voltage and the damage from excessively high open probability combine, causing the patch membrane to rupture. To address this issue, the CTIBD concentration was lowered to 3 μ M. In the single-channel recordings in Fig. 2, the membrane voltage level and open probability are very low, so the damage to the patch membrane is not significant, allowing the use of 10 μ M CTIBD.

7. Figure EV7 is very difficult to read. It might be an idea to make a new multi-panel figure in which the effect of each mutation can be clearly seen. At present it is very difficult to make out the effects of most mutations.

⇒ Figure EV7 was changed to the Figure S5. We have separated Figure S5 into individual panels for each mutant channel.

Minor Points

1. In Abstract, please rephrase the penultimate sentence to clearly state that CTIBD was less effective in the triple mutant. It currently suggests that CTIBD actually reduced the open probability of these channels, which is wrong.

⇒ We have replaced the sentence ' At the single-channel level, CTIBD treatment decreased the open-state probability in the triple mutant, mainly due to a drastically increased dissociation rate compared with the wild type.' in the abstract to the sentence ' At the single-channel level, CTIBD

treatment less affected the open-state probability in the triple mutant, mainly due to a drastically increased dissociation rate compared with the wild type.'

2. In the second sentence of the Introduction, insert "at physiological K⁺ gradients and membrane potentials" after "through the channel"

⇒ We have added "at physiological K⁺ gradients and membrane potentials" after "through the channel" in the second sentence of the introduction.

3. Please insert "not" into the sentence about the truncated BK channel (Bottom of 1st paragraph of Intro) so that it reads "the large intracellular domain are *NOT* activated by an increase"

⇒ We have inserted "not" into the sentence regarding the truncated BK channel (Bottom of 1st paragraph of Intro). Now, the sentence reads as follows: "Truncated BK_{Ca} channels lacking the large intracellular domain are not activated by an increase in Ca²⁺ because the RCK1 and RCK2 domains that sense intracellular Ca²⁺ are absent"

4. Introduction Paragraph 3, Last line and Paragraph 4, 6 lines from bottom. Please include references mentioning Cym04, BC5, omega-3 docosahexaenoic acid and the GoSlo family of BK channel openers since the molecular mechanisms of their mode of action is relevant.

⇒ We have mentioned cym04, BC5, omega-3, and the Goslo family at the end of the third paragraph of the introduction and provided references. The sentence is as follows: " Additionally, there are several BK_{Ca} channel activators such as Cym04, BC5, omega-3 docosahexaenoic acid, and the GoSlo family, which can be referred to understand the BK_{Ca} channel activation mechanism of CTIBD (Gessner G et al, 2012, Hoshi T et al, 2013, Webb Tl et al, 2015, Zhang G et al, 2022).".

5. Insert references in intro to the recently discovered LINGO family of BK regulatory subunits.

⇒ We have mentioned about LINGO family at the seventh line of the first paragraph of the introduction and provided the reference (Dudem S et al, 2020).

References

Dudem S, Large RJ, Kulkarni S, McClafferty H, Tikhonova IG, Sergeant GP, Thornbury KD, Shipston MJ, Perrino BA, Hollywood MA (2020) Lingo1 is a regulatory subunit of large conductance, ca²⁺-activated potassium channels. *Proceedings of the National Academy of Sciences* 117: 2194-2200.

Gessner G, Cui Y-M, Otani Y, Ohwada T, Soom M, Hoshi T, Heinemann SH (2012) Molecular mechanism of pharmacological activation of bk channels. *Proceedings of the National Academy of Sciences* 109: 3552-3557.

- Horrigan FT (2012) Conformational coupling in bk potassium channels. *Journal of General Physiology* 140: 625-634.
- Horrigan FT, Aldrich RW (2002) Coupling between voltage sensor activation, Ca^{2+} binding and channel opening in large conductance (bk) potassium channels. *The Journal of general physiology* 120: 267-305.
- Horrigan FT, Ma Z (2008) Mg^{2+} enhances voltage sensor/gate coupling in bk channels. *The Journal of general physiology* 131: 13-32.
- Hoshi T, Tian Y, Xu R, Heinemann SH, Hou S (2013) Mechanism of the modulation of bk potassium channel complexes with different auxiliary subunit compositions by the omega-3 fatty acid dha. *Proceedings of the National Academy of Sciences* 110: 4822-4827.
- Lee N, Lim BH, Lee K-S, Shin J, Pagire HS, Pagire SH, Ahn JH, Lee SW, Kang TM, Park C-S (2021) Identification and characterization of a novel large-conductance calcium-activated potassium channel activator, ctibd, and its relaxation effect on urinary bladder smooth muscle. *Molecular Pharmacology* 99: 114-124.
- Webb TI, Kshatri AS, Large RJ, Akande AM, Roy S, Sergeant GP, McHale NG, Thornbury KD, Hollywood MA (2015) Molecular mechanisms underlying the effect of the novel bk channel opener goslo: Involvement of the s4/s5 linker and the s6 segment. *Proceedings of the National Academy of Sciences* 112: 2064-2069.
- Zhang G, Xu X, Jia Z, Geng Y, Liang H, Shi J, Marras M, Abella C, Magleby KL, Silva JR (2022) An allosteric modulator activates bk channels by perturbing coupling between Ca^{2+} binding and pore opening. *Nature communications* 13: 6784.

June 18, 2024

Re: Life Science Alliance manuscript #LSA-2024-02621-TR

Prof. Chul-Seung Park
Gwangju Institute of Science and Technology
Life Sciences
Dept. of Life Science Gwanju Institute of Science & Technology 1 Oryng-dong Buk-gu
Gwang-ju, Buk-Gu 500-712
Korea, Republic of (South Korea)

Dear Dr. Park,

Thank you for submitting your revised manuscript entitled "Activation mechanism and novel binding sites of the BKCa channel activator CTIBD" to Life Science Alliance. The manuscript has been seen by the original reviewers whose comments are appended below. While the reviewers continue to be overall positive about the work in terms of its suitability for Life Science Alliance, some important issues remain.

Our general policy is that papers are considered through only one revision cycle; however, given that the suggested changes are relatively minor, we are open to one additional short round of revision. Please note that I will expect to make a final decision without additional reviewer input upon re-submission.

Please submit the final revision within one month, along with a letter that includes a point by point response to the remaining reviewer comments.

To upload the revised version of your manuscript, please log in to your account: <https://lsa.msubmit.net/cgi-bin/main.plex>
You will be guided to complete the submission of your revised manuscript and to fill in all necessary information.

B. MANUSCRIPT ORGANIZATION AND FORMATTING:

Sincerely,

Reviewer #1 (Comments to the Authors (Required)):

With one exception, the authors have provided reasonable rebuttals or further clarification of the issues raised in my initial review. My original point #2 still concerns me, but the issue may simply be a matter of semantics. Specifically, in my view, if a compound is shown to cause a shift in the $V_{1/2}$ of the G-V relationship, then by definition it has altered the voltage dependence of channel activation (in this case, shifting relationship to more negative membrane potentials). Altering voltage dependence can occur without a significant change in slope factor of the G-V relationship (as reported for many other drug-channel interactions). A shift in the G-V in no way indicates that the compound directly interacts with the voltage sensor - and this lack of interaction is well demonstrated by the compound-bound channel structure.

I suggest the authors simply add a sentence (either in Results or Discussion section) indicating that the voltage dependence of the channel activation is altered, but that other experimental evidence makes it clear that the compound does not physically interact with the canonical voltage sensor.

Reviewer #2 (Comments to the Authors (Required)):

The authors addressed my questions 1 and 2. The data of DEPC water injected oocytes should be included in Figure 1. For question 3 with regarding to the CTIBD2 site the response is not satisfactory. First, CTIBD in this site is also stabilized by CHS. It is not clear if the CTIBD2 site would be there in the absence of CHS. Second, the authors suggested that the CTIBD2 site might be responsible for a second phase of CTIBD effect. However, this suggestion is not supported by any data. On the other hand, the triple mutation seemed to eliminate most of the CTIBD effect, leaving little functional role for the CTIBD2 site. These structural and functional data need to be taken into the consideration on whether the CTIBD2 site is a true binding site.

Reviewer #3 (Comments to the Authors (Required)):

The authors have attempted to address a number of my previous concerns about the presentation and interpretation of their electrophysiology data. However, I still have a number of issues related to each of their replies.

1. The currents presented in the 3 μ M Ca²⁺ do not appear to be maximally activated, since the G/G_{max} provided in (A) of the Comments to the Author was >1 in the presence of the CTIBD. Given that the drug did not appear to alter single channel conductance (Fig 2 new manuscript), I find it hard to understand how G_{max} can increase above 1, if the currents are maximally activated in 1 μ M Ca²⁺. The authors need to clarify this in the revision. This also leads to a question concerning the buffers used to determine the [Ca²⁺] of each solution. The authors need to clarify in their methods exactly which Ca²⁺ buffer was used for each Ca²⁺ solution. Was it EGTA for 3 μ M Ca²⁺ or HEDTA?

2. Thank you for providing this information. However, the authors should include a few sentences in the discussion to at least recognise the possibility that the efficacy of the drug in 1 μ M Ca²⁺ may potentially be affected by changes in the activation $V_{1/2}$.

3. Good.

4. The authors need to state explicitly in their Methods that the currents were filtered at 1KHz post-acquisition.

5. The authors suggest that steady state currents can not be used due to "clamping and equipment specifications". Are they suggesting that the space clamp is inadequate, or that the voltage clamp is inadequate? This needs to be clarified since problems with voltage clamp means that the experimenter will not be able to reliably control the voltage during the depolarising step & therefore, the amplitude of the resultant tail current will be meaningless.

6. The problems with long continual depolarisations can be quite easily resolved by holding the patches repeatedly at the desired potential for 5 or 10s periods and resting them for 10s at more negative potentials.

7. This is a significant improvement on the previous MS and more clearly illustrates the effect of the each mutation. However, the authors need to carefully explain the limitations of presenting the data as G/G_{max} of vehicle. This is the same issue as described in point 1 above. If the channels are maximally activated and the drug does not significantly increase the single channel conductance, then the G/G_{max} should not increase about 1 in the presence of the drug. Where this is the case (eg Fig2C in drug and Fig S5, in drug with WT channels, W22A, F131A, W203A, S259A, P262A), there is a real concern here that either the $V_{1/2}$ in control or in drug is incorrect, as per my comments on the original manuscript. Alternatively, the drug may be affecting the single channel conductance. This potential limitation must be addressed in the discussion.

Minor Points

1. The penultimate sentence still lacks clarity. Please consider replacing it with something like "At the single channel level, CTIBD treatment was much less effective at increasing P_o in the triple mutant, mainly due to..."

2. OK.

3. OK.

4. OK, but if it is in keeping with Journal style, please order references so that they refer to the correct study ie Hoshi and Omega 3 etc.

5. OK.

On reading the reply to Reviewer 2 comment 2 the authors incorrectly state that the Webb et al., study utilised rat BK channels. They did not. They used rabbit BK.

Point-by-point response to reviewers

Reviewer #1 (Comments to the Authors (Required)):

With one exception, the authors have provided reasonable rebuttals or further clarification of the issues raised in my initial review. My original point #2 still concerns me, but the issue may simply be a matter of semantics. Specifically, in my view, if a compound is shown to cause a shift in the $V_{1/2}$ of the G-V relationship, then by definition it has altered the voltage dependence of channel activation (in this case, shifting relationship to more negative membrane potentials). Altering voltage dependence can occur without a significant change in slope factor of the G-V relationship (as reported for many other drug-channel interactions). A shift in the G-V in no way indicates that the compound directly interacts with the voltage sensor - and this lack of interaction is well demonstrated by the compound-bound channel structure.

I suggest the authors simply add a sentence (either in Results or Discussion section) indicating that the voltage dependence of the channel activation is altered, but that other experimental evidence makes it clear that the compound does not physically interact with the canonical voltage sensor.

⇒ As the reviewer suggested, we have added the following sentence to the first paragraph of the discussion based on the comment: 'Although the $V_{1/2}$, which represents voltage dependence of the channel activation is altered by CTIBD (Fig 4), the open probability within -90 to -60 mV (Fig 2) and the cryo-EM structure (Fig 3) make it clear that the compound does not physically interact with the voltage sensor.' This sentence is highlighted in red.

Reviewer #2 (Comments to the Authors (Required)):

The authors addressed my questions 1 and 2. The data of DEPC water injected oocytes should be included in Figure 1.

⇒ We added the representative current trace from the oocytes injected with DEPC water as a negative control in Fig 1B. In order to explain the new figure, we have added the following sentences to the first paragraph of the 'Ca²⁺ and Voltage Sensing Are Not Essential for CTIBD-mediated BK_{Ca} Channel Activation' section of the results: 'When only the nuclease-free water without mRNA was injected into the oocytes, small leak currents were observed (Fig 1B). Based on this fact, it can be inferred that the currents recorded from oocytes injected with *rSlo1-Kv-minT* mRNA were due to the opening of the modified channels expressed on the

membrane.' This part is highlighted in red.

For question 3 with regarding to the CTIBD2 site the response is not satisfactory. First, CTIBD in this site is also stabilized by CHS. It is not clear if the CTIBD2 site would be there in the absence of CHS. Second, the authors suggested that the CTIBD2 site might be responsible for a second phase of CTIBD effect. However, this suggestion is not supported by any data. On the other hand, the triple mutation seemed to eliminate most of the CTIBD effect, leaving little functional role for the CTIBD2 site. These structural and functional data need to be taken into the consideration on whether the CTIBD2 site is a true binding site.

⇒ In response to the reviewer's concerns, we have revised the manuscript to state that although our structure reveals two different binding sites for CTIBD, we do not rule out the possibility that the CTIBD2 binding site may be non-physiological, potentially caused by the high concentration of CTIBD used in grid preparation. The fact that the triple mutation eliminates most of the CTIBD effect further supports this hypothesis. We have added the following sentences to the second paragraph of the discussion: 'One possible explanation for this unexpected result is that the CTIBD2 binding site may not function as a binding site under physiological conditions. The CTIBD2 binding site is observed under conditions treated with high concentrations of CTIBD using cryo-EM and found to be stabilized in the presence of CHS. In addition, the effects of CTIBD are almost negligible in the triple mutant channel further supporting the possibility that the occupancy of CTIBD to this second site is artifactual under experimental conditions or not physiologically functional at least.' This sentence is highlighted in red.

Reviewer #3 (Comments to the Authors (Required)):

The authors have attempted to address a number of my previous concerns about the presentation and interpretation of their electrophysiology data. However, I still have a number of issues related to each of their replies.

1. The currents presented in the 3 μM Ca^{2+} do not appear to be maximally activated, since the G/G_{max} provided in (A) of the Comments to the Author was >1 in the presence of the CTIBD. Given that the drug did not appear to alter single channel conductance (Fig 2 new manuscript), I find it hard to understand how G_{max} can increase above 1, if the currents are maximally activated in 1 μM Ca^{2+} . The authors need to clarify this in the revision. This also leads to a question concerning the buffers used to determine the $[\text{Ca}^{2+}]$ of each solution. The authors

need to clarify in their methods exactly which Ca²⁺ buffer was used for each Ca²⁺ solution. Was it EGTA for 3 μ M Ca²⁺ or HEDTA?

⇒ We agree with the reviewer's comment. It is difficult to understand how G_{max} changes with CTIBD treatment while single-channel conductance remains unchanged. One possible explanation for this phenomenon is that single-channel conductance might change due to CTIBD at very high membrane voltages or CTIBD concentrations. In this paper, single-channel conductance was measured only within the range of 20 to 60 mV and 3 μ M CTIBD concentration, because the patch membrane easily ruptures at higher membrane voltages or CTIBD concentrations under the single-channel recording protocol. Therefore, it is still possible that single-channel conductance could be affected by CTIBD at membrane voltages above 60 mV and concentrations exceeding 3 μ M of CTIBD. Several previous studies reported that some BK_{Ca} channel modulators can alter the single-channel conductance of the channel (Monat J et al, 2024, Moss BL & Magleby KL, 2001, Scornik FS et al, 2013). This could explain the changes in G_{max} caused by CTIBD. To clarify this for readers, we have added the following sentences to the middle of the 'Single-channel Recording of WT and Triple-mutant BK_{Ca} Channels with CTIBD treatment' section of the results: 'The single-channel conductances were measured within the range of 20 to 60 mV and at a CTIBD concentration of 3 μ M. Above 60 mV or at concentrations exceeding 3 μ M of CTIBD, there remains the possibility that CTIBD can alter the single-channel conductance and G_{max} (Figs 4C and S5).' This part is highlighted in red.

Due to the limitations of outside-out recording (inability to change the internal calcium solution of the pipette during experiments), we normalized the conductance of WT and the triple mutant for each patch membrane separately in the Fig 2 data. Normalization was conducted based on the G_{max} of each patch membrane, resulting in a maximum value of 1 for G/G_{max} . However, in Figs 4C and S5 of the main manuscript, we treated the vehicle on the outside of the patch membrane and then switched to CTIBD with the same patch membrane. The conductances of the vehicle-treated and CTIBD-treated conditions obtained on the same patch membrane could be normalized by the G_{max} values obtained from the vehicle-treated condition. Therefore, as explained above, if single-channel conductance changes under membrane potentials exceeding 60 mV and CTIBD concentrations exceeding 3 μ M, G_{max} may increase compared to the vehicle-treated condition. Thus, normalization of the CTIBD-treated condition conductance using G_{max} from the vehicle-treated condition could result in values above 1.

We used EGTA to prepare calcium solutions of 1 μ M or lower and HEDTA to prepare calcium solutions greater than 1 μ M (Patton C et al, 2004). In order to specify this, we have added the following sentence to the 'Electrophysiological Recordings and Data Analysis' section of the

Materials and Methods: 'HEDTA was used to prepare calcium solutions greater than 1 μM , while EGTA was used to prepare calcium solutions of 1 μM or lower (Patton C et al, 2004).' This sentence is highlighted in red.

2. *Thank you for providing this information. However, the authors should include a few sentences in the discussion to at least recognise the possibility that the efficacy of the drug in 1 μM Ca^{2+} may potentially be affected by changes in the activation $V_{1/2}$.*

⇒ We have added and revised the following sentences to the third paragraph of the discussion based on your comment: 'All experiments in this paper were conducted under 3 μM or 10 μM Ca^{2+} concentrations. Therefore, it is possible that the efficacy of CTIBD could be influenced by changes in the activation $V_{1/2}$ under different Ca^{2+} concentrations. Even so, these results strongly indicate that these three residues are crucial, if not essential, for the binding and activation of CTIBD.' This part is highlighted in red.

3. *Good.*

4. *The authors need to state explicitly in their Methods that the currents were filtered at 1KHz post-acquisition.*

⇒ We have added 'post-acquisition' to the sentence in the 'Electrophysiological Recordings and Data Analysis' section of the Materials and Methods: 'To reduce the noise, single-channel recording data were low-pass filtered at 1 kHz post-acquisition with an eight-pole Bessel filter using Clampfit 11.0.3 software.' The modified section is highlighted in red.

5. *The authors suggest that steady state currents can not be used due to "clamping and equipment specifications". Are they suggesting that the space clamp is inadequate, or that the voltage clamp is inadequate? This needs to be clarified since problems with voltage clamp means that the experimenter will not be able to reliably control the voltage during the depolarising step & therefore, the amplitude of the resultant tail current will be meaningless.*

⇒ We apologize for the inaccuracies and lack of detail in my previous response. BK_{Ca} channels could be blocked by various types of cations that are present in pipette solution or the oocytes in a voltage-dependent manner (Brelidze TI & Magleby KL, 2004, Díez-Sampedro A et al, 2006, Zhang Y et al, 2006). Therefore, BK_{Ca} channels permeating outward K^+ currents in the patch membrane are blocked by intracellular divalent cations such as Ca^{2+} especially

under highly positive voltages. Consequently, the current fails to reach the expected level from the number of open channels, leading to a slow flattening phenomenon in which the current decreases slightly. We used the term "clamping and equipment specifications" because this slow flattening phenomenon prevents the current level from reaching what should be produced by the actual number of open channels, and it cannot be resolved by changing the equipment specifications.

When analyzing steady-state current, one must consider not only slow flattening phenomenon but also the changes in driving force and leak conductance according to the voltage level. However, in the case of tail current, since it is obtained at a single voltage level (-100 mV), there is no difference in the driving force or leak conductance for all data points. Additionally, the electrogradient of tail current is opposite to the positive membrane voltage, making it free from cation block. For these reasons, analyzing tail current yields more accurate results, and many papers related to BK_{Ca} channels analyze data using tail current (Gonzalez-Perez V et al, 2014, Pusch M, 2006, Zhang G et al, 2022).

6. The problems with long continual depolarisations can be quite easily resolved by holding the patches repeatedly at the desired potential for 5 or 10s periods and resting them for 10s at more negative potentials.

⇒ We appreciate the kind comment. During single-channel recording at a 10 μM CTIBD concentration, the patch membrane ruptured before sufficient data could be obtained, usually within one or two seconds. In future research where single-channel recording data should be obtained at a high CTIBD concentration, we will refer to this comment to establish an appropriate experimental protocol.

7. This is a significant improvement on the previous MS and more clearly illustrates the effect of the each mutation. However, the authors need to carefully explain the limitations of presenting the data as G/Gmax of vehicle. This is the same issue as described in point 1 above. If the channels are maximally activated and the drug does not significantly increase the single channel conductance, then the G/Gmax should not increase about 1 in the presence of the drug. Where this is the case (eg Fig2C in drug and Fig S5, in drug with WT channels, W22A, F131A, W203A, S259A, P262A), there is a real concern here that either the V1/2 in control or in drug is incorrect, as per my comments on the original manuscript. Alternatively, the drug may be affecting the single channel conductance. This potential limitation must be addressed in the discussion.

⇒ We believe that the response to Comment 1 above is sufficient as an answer to this comment.

Minor Points

1. The penultimate sentence still lacks clarity. Please consider replacing it with something like "At the single channel level, CTIBD treatment was much less effective at increasing P_o in the triple mutant, mainly due to..."

⇒ We have revised the corresponding section of the abstract as follows: 'At the single channel level, CTIBD treatment was much less effective at increasing P_o in the triple mutant, mainly due to a drastically increased dissociation rate compared with the wild type.' This sentence is highlighted in red.

2. OK.

3. OK.

4. OK, but if it is in keeping with Journal style, please order references so that they refer to the correct study ie Hoshi and Omega 3 etc.

⇒ We have revised the order of BK channel openers in the third paragraph of the introduction section. The revised sentence is as follows: 'Additionally, there are several BK_{Ca} channel activators such as Cym04, omega-3 docosahexaenoic acid, GoSlo family, and BC5, which can be referred to understand the BK_{Ca} channel activation mechanism of CTIBD.' The revised part is highlighted in red.

5. OK.

On reading the reply to Reviewer 2 comment 2 the authors incorrectly state that the Webb et al., study utilised rat BK channels. They did not. They used rabbit BK.

⇒ Thank you for the accurate correction. We confused the meaning of rSlo in the paper. However, as mentioned in the previous response, since Horrigan's paper used the mouse BK channel, it remains valid to suggest that there may be different values for open probability in this paper using the rat BK channel.

References

- Brelidze TI, Magleby KL (2004) Protons block bk channels by competitive inhibition with k⁺ and contribute to the limits of unitary currents at high voltages. *The Journal of general physiology* 123: 305-319.
- Díez-Sampedro A, Silverman WR, Bautista JF, Richerson GB (2006) Mechanism of increased open probability by a mutation of the bk channel. *Journal of neurophysiology* 96: 1507-1516.
- Gonzalez-Perez V, Xia X-M, Lingle CJ (2014) Functional regulation of bk potassium channels by γ 1 auxiliary subunits. *Proceedings of the National Academy of Sciences* 111: 4868-4873.
- Monat J, Altieri LG, Enrique N, Sedán D, Andrinolo D, Milesi V, Martín P (2024) Direct inhibition of bk channels by cannabidiol, one of the principal therapeutic cannabinoids derived from cannabis sativa. *Journal of Natural Products*:
- Moss BL, Magleby KL (2001) Gating and conductance properties of bk channels are modulated by the s9–s10 tail domain of the α subunit: A study of mslo1 and mslo3 wild-type and chimeric channels. *The Journal of general physiology* 118: 711-734.
- Pusch M (2006) Analysis of electrophysiological data. *Expression and analysis of recombinant ion channels: from structural studies to pharmacological screening*: 111-144.
- Scornik FS, Bucciero RS, Wu Y, Selga E, Bosch Calero C, Brugada R, Pérez GJ (2013) Dibac4 (3) hits a “sweet spot” for the activation of arterial large-conductance ca²⁺-activated potassium channels independently of the β 1-subunit. *American Journal of Physiology-Heart and Circulatory Physiology* 304: H1471-H1482.
- Zhang G, Xu X, Jia Z, Geng Y, Liang H, Shi J, Marras M, Abella C, Magleby KL, Silva JR (2022) An allosteric modulator activates bk channels by perturbing coupling between ca²⁺ binding and pore opening. *Nature communications* 13: 6784.
- Zhang Y, Niu X, Brelidze TI, Magleby KL (2006) Ring of negative charge in bk channels facilitates block by intracellular mg²⁺ and polyamines through electrostatics. *The Journal of general physiology* 128: 185-202.

July 8, 2024

RE: Life Science Alliance Manuscript #LSA-2024-02621-TRR

Prof. Chul-Seung Park
Gwangju Institute of Science and Technology
Life Sciences
Dept. of Life Science Gwanju Institute of Science & Technology 1 Oryng-dong Buk-gu
Gwang-ju, Buk-Gu 500-712
Korea, Republic of (South Korea)

Dear Dr. Park,

Thank you for submitting your revised manuscript entitled "Activation mechanism and novel binding sites of the BKCa channel activator CTIBD". We would be happy to publish your paper in Life Science Alliance pending final revisions necessary to meet our formatting guidelines.

- please be sure that the authorship listing and order is correct
- please upload a clean manuscript file without highlights/tracked-changes
- please add callouts for Figures S1A-C; S3A-C; S6A-B; S7A-B and S8A-D to your main manuscript text

A. FINAL FILES:

B. MANUSCRIPT ORGANIZATION AND FORMATTING:

Sincerely,

July 15, 2024

RE: Life Science Alliance Manuscript #LSA-2024-02621-TRRR

Prof. Chul-Seung Park
Gwangju Institute of Science and Technology
Life Sciences
Dept. of Life Science Gwanju Institute of Science & Technology 1 Oryng-dong Buk-gu
Gwang-ju, Buk-Gu 500-712
Korea, Republic of (South Korea)

Dear Dr. Park,

Thank you for submitting your Research Article entitled "Activation mechanism and novel binding sites of the BKCa channel activator CTIBD". It is a pleasure to let you know that your manuscript is now accepted for publication in Life Science Alliance. Congratulations on this interesting work.

DISTRIBUTION OF MATERIALS:

Again, congratulations on a very nice paper. I hope you found the review process to be constructive and are pleased with how the manuscript was handled editorially. We look forward to future exciting submissions from your lab.

Sincerely,
